# MoMA: Model-based Mirror Ascent for Offline Reinforcement Learning

**Mao Hong**                                          *mhong26@jhu.edu*
*Department of Applied Mathematics and Statistics*
*Johns Hopkins University*

**Zhiyue Zhang**                                      *zzhan179@jhu.edu*
*Department of Applied Mathematics and Statistics*
*Johns Hopkins University*

**Yue Wu**                                            *ywu166@jhu.edu*
*Department of Applied Mathematics and Statistics*
*Johns Hopkins University*

**Yanxun Xu**                                         *yanxun.xu@jhu.edu*
*Department of Applied Mathematics and Statistics*
*Division of Quantitative Sciences*
*Johns Hopkins University*

**Reviewed on OpenReview:** *https://openreview.net/forum?id=RHUKg8n9tw*

## Abstract

Model-based offline reinforcement learning methods (RL) have achieved state-of-the-art performance in many decision-making problems thanks to their sample efficiency and generalizability. Despite these advancements, existing model-based offline RL approaches either focus on theoretical studies without developing practical algorithms or rely on a restricted parametric policy space, thus not fully leveraging the advantages of an unrestricted policy space inherent to model-based methods. To address this limitation, we develop MoMA, a model-based mirror ascent algorithm with general function approximations under partial coverage of offline data. MoMA distinguishes itself from existing literature by employing an unrestricted policy class. In each iteration, MoMA conservatively estimates the value function by a minimization procedure within a confidence set of transition models in the policy evaluation step, then updates the policy with general function approximations instead of commonly-used parametric policy classes in the policy improvement step. Under some mild assumptions, we establish theoretical guarantees for MoMA by proving an upper bound on the suboptimality of the returned policy. We also provide a practically implementable, approximate version of the algorithm. The effectiveness of MoMA is demonstrated via numerical studies.

## 1 Introduction

Reinforcement Learning (RL) has emerged as an effective approach for optimizing sequential decision making by maximizing the expected cumulative reward to learn an optimal policy through iterative online interactions with the environment. RL algorithms have made significant advances in a wide range of areas such as autonomous driving (Shalev-Shwartz et al., 2016), video games (Torrado et al., 2018), and robotics (Kober et al., 2013). However, numerous real-world problems require methods to learn only from pre-collected and static (i.e., offline) datasets because interacting with the environment can be expensive or unethical, such as assigning patients to inferior or toxic treatments in healthcare applications (Gottesman et al., 2019).

Therefore, the development of offline RL methods, which learn an optimal policy solely from offline data without further interactions with the environment, has grown rapidly in recent decades (Levine et al., 2020).

The performance of offline RL methods often relies on the coverage of offline data. Earlier theoretical studies of offline RL usually assume that offline data have full coverage, i.e., every possible policy's occupancy measure can be covered by the occupancy measure of the behavior policy that generates offline data (Munos, 2003; Antos et al., 2008; Munos & Szepesvári, 2008; Farahmand et al., 2010; Lange et al., 2012; Ross & Bagnell, 2012; Chen & Jiang, 2019; Liu et al., 2019; Uehara et al., 2020; Xie & Jiang, 2020; 2021). This assumption implies a highly exploratory behavior policy that can explore all state-action pairs, a condition rarely met in practical scenarios. Recent studies have shifted from this restrictive full coverage assumption to a more realistic partial coverage framework. This only requires the behavior policy's occupancy measure to cover that of the target policy. Methods developed under the partial coverage assumption fall into two main categories: policy constraint methods and pessimistic methods. Policy constraint methods focus on learning an optimal policy within a boundary that maintains proximity to the behavior policy (Fujimoto et al., 2019; Kumar et al., 2019; Wu et al., 2019; Liu et al., 2019; Laroche et al., 2019; Nachum et al., 2019; Siegel et al., 2020; Kostrikov et al., 2021; Fujimoto & Gu, 2021), ensuring that the performance of the learned policy is at least as good as the behavior policy. On the other hand, pessimistic methods, which include both model-free (Kumar et al., 2020; Liu et al., 2020; Jin et al., 2021; Rashidinejad et al., 2021; Xie et al., 2021; Zanette et al., 2021; Yin & Wang, 2021; Kostrikov et al., 2021; Cheng et al., 2022; Shi et al., 2022; Zhang et al., 2022) and model-based approaches (Yu et al., 2020; Kidambi et al., 2020; Chang et al., 2021; Uehara & Sun, 2021; Yu et al., 2021; Rigter et al., 2022; Guo et al., 2022; Rashidinejad et al., 2022; Bhardwaj et al., 2023), employ a principle of pessimism to penalize less-visited state-action pairs without constraining the policy, thereby avoiding uncertain regions not covered by the offline data.

Compared to pessimistic model-free methods, a key advantage of pessimistic model-based methods lies in their ability to utilize an unrestricted policy space (Uehara & Sun, 2021). Nonetheless, existing offline model-based approaches (Uehara & Sun, 2021; Rigter et al., 2022; Guo et al., 2022; Rashidinejad et al., 2022; Bhardwaj et al., 2023) either focus on theoretical studies without developing practical algorithms, or rely on a restricted parametric policy space, thereby not fully exploiting the potential of an unrestricted policy space inherent to model-based methods. The drawback of using parameterized policy classes becomes evident when the optimal policy falls outside the predefined policy class. To circumvent the constraint of parametric policy spaces, this paper aims to develop a pessimistic model-based algorithm that employs an unrestricted policy class.

In this study, we introduce MoMA, a pessimistic model-based mirror ascent algorithm designed for offline RL, without the need for explicit policy parameterization. At a high level, MoMA iteratively performs two steps: conservative policy evaluation and policy improvement. In the conservative policy evaluation step, we find a pessimistic $Q$ function of the current policy at each iteration $t$ by minimizing the $Q$ function over a confidence set of transition models. It plays a critical role in relaxing the full coverage assumption to partial coverage in offline RL. In the policy improvement step, given the pessimistic $Q$ function from the first step, we update the policy by mirror ascent (Beck & Teboulle, 2003) with general function approximations. The design of the policy improvement step is adapted from Lan (2022) to facilitate an unrestricted policy class in our model-based offline RL setting.

The theoretical framework of MoMA separates conservative policy evaluation from policy improvement. This separation is advantageous as it allows for independent investigation of statistical and computational complexities. Under this theoretical framework, we establish an upper bound on the suboptimality of MoMA with a characterization of statistical error, optimization error, and function approximation error, assuming a computational oracle for a constrained minimization problem in the conservative policy evaluation step. A notable distinction of our work, compared to model-free approaches like Xie et al. (2021), is that our suboptimality upper bound does not include the size of the policy class. This is a significant advantage, as it permits the assumption of an unrestricted policy class. In model-free settings, the policy-dependent confidence set must maintain certain properties uniformly across all policies in the class, resulting in the inclusion of the policy class size in the suboptimality upper bound. In contrast, in our model-based approach, the confidence set is independent of the policy. This independence eliminates the need to factor in the size of the policy class, thereby removing constraints on its expansiveness. On the computational side, we develop

a practically implementable algorithm that serves as an approximation of the theoretical algorithm. In particular, a primal-dual step is designed to address the computational oracle requirement - approximately solving the constrained minimization problem in the conservative policy evaluation step.

The rest of the paper is organized as follows. Section 2 introduces some preliminary settings and definitions. In Section 3, we present MoMA, the proposed algorithm, followed by Section 4, which discusses its practical, approximate version suitable for implementation. Section 5 establishes the main theoretical results for MoMA under some assumptions. Numerical results in both synthetic dataset and D4RL benchmark are included in section 6. Section 7 discusses the comparison between MoMA and several existing works in offline RL. Finally, Section 8 concludes the paper with a discussion.

## 2 Preliminaries

**Markov decision processes and offline RL:** We consider an infinite-horizon Markov decision process (MDP) $M = (\mathcal{S}, \mathcal{A}, P, r, \gamma, \mu_0)$, with continuous state space $\mathcal{S}$, discrete action space $\mathcal{A} = \{A_1, A_2, ..., A_m\}$, a transition dynamics $P(s' \mid s, a)$ with $s, s' \in \mathcal{S}$ and $a \in \mathcal{A}$, a reward function $r : \mathcal{S} \times \mathcal{A} \to [0, 1]$, a discount factor $\gamma \in [0, 1)$, and an initial state distribution $\mu_0$. We assume a model space $\mathcal{P}$ for the transition dynamic, i.e. $P \in \mathcal{P}$. The reward function $r$ is assumed to be known throughout this work. A stochastic policy $\pi$ maps from state space to a distribution over actions, representing a decision strategy to pick an action with probability $\pi(\cdot|s)$ given the current state $s$, i.e. $\pi(\cdot \mid s) \in \Delta(\mathcal{A}) := \{p \in \mathbb{R}^m : \sum_{i=1}^m p_i = 1, p_i \geq 0, \forall i\}$ for all $s \in \mathcal{S}$. Given a policy $\pi$ and a transition dynamics $P$, the value function $V_P^\pi(s) := \mathbb{E}_{P,\pi}[\sum_{t=0}^\infty \gamma^t r(s_t, a_t)|s_0 = s]$ denotes the expected cumulative discounted reward of $\pi$ under the transition dynamics $P$ with an initial state $s$ and a reward function $r$. We use $V_P^\pi := \mathbb{E}_{s \sim \mu_0} V_P^\pi(s)$ to denote the expected value integrated over $\mathcal{S}$ with an initial distribution $\mu_0$. The action-value function (i.e., $Q$ function) is defined similarly: $Q_P^\pi(s, a) = \mathbb{E}_{P,\pi}[\sum_{t=0}^\infty \gamma^t r(s_t, a_t)|s_0 = s, a_0 = a]$. Let $d_P^\pi(s, a) := (1-\gamma) \sum_{t=0}^\infty \gamma^t \Pr(s_t = s, a_t = a|s_0 \sim \mu_0)$ be the occupancy measure of the policy $\pi$ under the dynamics $P$. Then $V_P^\pi$ can be expressed as $\mathbb{E}_{(s,a) \sim d_P^\pi}[r(s, a)]$. Assuming that a static offline dataset $\mathcal{D}_n = \{(s_i, a_i, r_i, s_i') : i = 1, ..., n\}$ is generated by some behavior policy under the ground truth transition dynamics $P^*$, model-based offline RL methods aim to learn an optimal policy that maximizes the value $V_{P^*}^\pi$ through learning the dynamics from the offline dataset without any further interactions with the environment.

**Partial coverage:** One fundamental challenge in offline RL is distribution shift (Levine et al., 2020): the visitation distribution of states and actions induced by the learned policy inevitably deviates from the distribution of offline data. The concept of coverage has been introduced to measure the distribution shift using the density ratio (Chen & Jiang, 2019). Denote $\rho(s, a)$ to be the offline distribution that generates the state-action pairs $(s_i, a_i)_{i=1}^n$ in offline data. Full coverage means $\sup_{s,a} d_{P^*}^\pi(s, a)/\rho(s, a) < \infty$ for all possible policies $\pi$, which may not hold in practice. In contrast, partial coverage only assumes that the offline distribution covers the visitation distribution induced by some comparator policy $\pi^\dagger$ (Xie et al., 2021), such that $\sup_{s,a} d_{P^*}^{\pi^\dagger}(s, a)/\rho(s, a) < \infty$. Our work aims to learn the optimal policy among all polices covered by offline data, i.e., $\Pi := \{\pi : \sup_{s,a} d_{p^*}^\pi(s, a)/\rho(s, a) < \infty\}$.

## 3 Model-based mirror ascent for offline RL

In this section, we present MoMA, which is summarized in Algorithm 1. MoMA can be separated into two steps in each iteration: 1) In the policy evaluation step, we conservatively evaluate the updated policy through a minimization procedure within a confidence set of transition models; 2) In the policy improvement step, we update the policy based on mirror ascent (MA) under the current transition model. The conservative policy optimization procedure is designed to mitigate distribution shift by penalizing the value of state-action pairs that are rarely visited, thereby addressing the high uncertainty in their value estimation. This approach ensures performance under a partial coverage assumption rather than a full coverage assumption. We provide details for the policy evaluation and policy improvement in section 3.1 and section 3.2 respectively.

---

**Algorithm 1** MoMA: Model-based mirror ascent for offline RL

---

**Input:** The learning rate $\{\eta_t\}_{t=1}^T$, a consistent estimate $\widehat{P}$ and its corresponding $\alpha_n$.
**Initialization:** Initialize $\pi_0(\cdot|s) = \text{Unif}(\mathcal{A})$.
**for** $t = 1$ **to** $T$ **do**
    Conservative policy evaluation:
    Let $P_t = \text{argmin}_{P \in \mathcal{P}_{n,\alpha_n}} V_P^{\pi_t}$, where $\mathcal{P}_{n,\alpha_n} = \{P \in \mathcal{P} : \mathcal{E}_n(P) \le \alpha_n\}$.
    Policy improvement: $\pi_{t+1}(\cdot \mid s) = \underset{p \in \Delta(\mathcal{A})}{\arg\max} \left\{ \langle Q_{P_t}^{\pi_t}(s, \cdot), p \rangle - \frac{1}{\eta_t} D\left(\pi_t(\cdot \mid s), p\right) \right\}, \forall s.$
**end for**

---

### 3.1 Policy evaluation: conservative estimate of $Q$

In the policy evaluation step, inspired by Uehara & Sun (2021), we first construct a confidence set

$$\mathcal{P}_{n,\alpha_n} = \{P \in \mathcal{P} : \mathcal{E}_n(P) \le \alpha_n\}$$

for transition models. Here

$$\mathcal{E}_n(P) := \widehat{L}_n(P) - \widehat{L}_n(\widehat{P}),$$

where $\widehat{L}_n : \mathcal{P} \to \mathbb{R}^+$ is an empirical loss function for $P$, depending on the offline dataset $\mathcal{D}_n$, and

$$\widehat{P} = \arg\min_{P \in \mathcal{P}} \widehat{L}_n(P)$$

is an estimator based on $\widehat{L}_n$. For example, if $\widehat{L}_n(P)$ denotes the negative log-likelihood function of $P$, then $\widehat{P}$ is the maximum likelihood estimator (MLE) for $P$. $\alpha_n$ can be understood as the radius of the confidence set. Then, we find $P_t$ that minimizes the value function $V_P^{\pi_t}$ within $\mathcal{P}_{n,\alpha_n}$, which is formulated as

$$P_t = \text{argmin}_{P \in \mathcal{P}_{n,\alpha_n}} V_P^{\pi_t}. \tag{1}$$

If the radius of the confidence set $\alpha_n$ is set to zero, then $P_t$ exactly corresponds to the MLE for $P$. In this scenario, no uncertainty penalization is considered. A positive $\alpha_n$ allows us to construct a perturbation around the MLE. Finding the most pessimistic $P_t$ within the confidence region implicitly imposes an uncertainty penalization for rarely visited state-action pairs.

The minimizer $P_t$, combined with the current policy $\pi_t$, can be used to evaluate $Q_{P_t}^{\pi_t}$ through Monte Carlo methods. In section 4.3, we will provide an example using negative log-likelihood to illustrate the implementation procedure.

The conservative policy evaluation step is designed to provide a pessimistic estimate of the value function given a policy. This idea has been employed in pessimistic model-free actor-critic algorithms (Khodadian et al., 2021; Zanette et al., 2021; Xie et al., 2021), where the critic lower bounds the $Q$ function. For example, Khodadadian et al. (2021) constructed the pessimism for $Q$ under the tabular setting. Zanette et al. (2021) assumed a linear $Q$ function and conservatively estimated $Q$ by minimizing $Q$ within a confidence set of the coefficients for $Q$. Xie et al. (2021) conservatively estimated $Q$ in a general function approximation setting, however, the size of the function class was limited by the sample size of offline data. Compared to these model-free methods, MoMA has several advantages. First, unlike Xie et al. (2021), the size of the function class for approximation in MoMA can be arbitrarily large. Second, MoMA has no restriction on the policy class, which is crucial when the optimal policy is not contained in a restricted parametric policy class. See Appendix 7 for a detailed discussion about the comparisons between our work and existing literature in the policy evaluation step.

### 3.2 Policy improvement: mirror ascent

In the policy improvement step, we use mirror ascent, which maximizes the $Q$ function with a regularizer that penalizes the Bregman distance $D(\cdot, \cdot)$ between the next policy and the current policy. We first introduce

---
**Algorithm 2** MoMA: A Practical Algorithm
___
    **Input:** The learning rate $\eta_t$, $\mathcal{P}_{n,\alpha_n}$.
    **Initialization:** Initialize $\pi_0 = \text{Unif}(\mathcal{A})$.
    **for** $t = 1$ **to** $T$ **do**
        Conservative policy evaluation:
        Compute $P_t := P_{\phi^{(K)}}$, where $\phi^{(K)}$ is the output from eq. (4).
        Policy improvement:
        Sample $\{s_j\}_{j=1}^N$ from $d_{P_t}^{\pi_t}$.
        **for** $j = 1$ **to** $N$ **do**
            Input $\{f_{t-1,i}(s; \widehat{\beta}_{t-1,i})\}_{i=1}^m$ and $s_j$ into algorithm 4, and output $\{\tilde{Q}_{\omega,t}(s_j, A_i)\}_{i=1}^m$.
        **end for**
        Find $\widehat{\beta}_{t,i}$ that solves eq. (6) for each $i = 1, ..., m$.
        Save the parametric function $\{f_{t,i}(s; \widehat{\beta}_{t,i})\}_{i=1}^m$ as an input for iteration $t + 1$.
    **end for**
___

the definition of the Bregman distance. Let $\| \cdot \|$ be a given norm in $\Delta(\mathcal{A})$ and $\omega : \Delta(\mathcal{A}) \to \mathbb{R}$ be a strongly convex function with respect to (w.r.t.) $\| \cdot \|$. Then $D(\cdot, \cdot)$ is a Bregman distance if

$$D(p, p\prime) := \omega(p\prime) - [\omega(p) + \langle \nabla \omega(p), p\prime - p \rangle] \geq \frac{1}{2} \|p\prime - p\|^2, \forall p, p\prime \in \Delta(\mathcal{A}),$$

where $\langle \cdot, \cdot \rangle$ denotes the inner product. For clarity, we consider $Q_{P_t}^{\pi_t}(s, \cdot) \in \mathbb{R}^m$ with $i$-th element $Q_{P_t}^{\pi_t}(s, A_i)$. Given a pre-specified learning rate $\eta_t > 0$, the proposed update rule is:

$$\pi_{t+1}(\cdot \mid s) = \underset{p \in \Delta(\mathcal{A})}{\arg\max} \left\{ \langle Q_{P_t}^{\pi_t}(s, \cdot), p \rangle - \frac{1}{\eta_t} D(\pi_t(\cdot \mid s), p) \right\} \quad \forall s. \tag{2}$$

Intuitively, the update rule aims to maximize the value function while ensuring minimal deviation from the previous step. The distance $D$ measures the divergence between the updated policy and the original policy from the previous step. The parameter $\eta_t$ can be interpreted as the step size, indicating the weight placed on distance control. This update rule is distinct from existing literature in that it does not require any explicit policy parameterization. This feature underscores the advantage of MoMA over existing model-based offline algorithms that rely on parametric policy classes, e.g., Rigter et al. (2022); Guo et al. (2022); Rashidinejad et al. (2022); Bhardwaj et al. (2023).

As the Bregman distance defines a general class of distance measures, one can design corresponding algorithms based on specific distance measures if desired. Notably, natural policy gradient (NPG) (Kakade, 2001) is a special case of policy mirror ascent when $D(\cdot, \cdot)$ is set to be KL divergence.

We remark that for a continuous state space $S$, it is impossible to enumerate (2) for infinitely many states. To overcome this issue, in section 4.2 we provide a computationally efficient algorithm through function approximation, which is one of the key advantages of MoMA compared to existing literature, e.g., Algorithm 1 in Xie et al. (2021).

## 4 A practical algorithm

In this section, we present an implementable algorithm that approximately solves the constrained minimization problem (1), summarized in algorithm 2.

### 4.1 Primal-dual (PD) for solving the constrained optimization problem

In order to approximately solve the constrained minimization problem (1) in the policy evaluation step, we introduce its Lagrangian form:

$$\max_{\lambda > 0} \min_{P \in \mathcal{P}} V_P^{\pi_t} + \lambda(\mathcal{E}_n(P) - \alpha_n). \tag{3}$$

Suppose the transition model $P$ is parameterized by $\phi$, denoted as $P_\phi$. To solve (3), we design a model gradient primal-dual method with the following update rule:

$$\phi^{(k+1)} = \phi^{(k)} - \kappa_1 \nabla_\phi \mathcal{L}_{V,t}(\phi^{(k)}, \lambda^{(k)}), \quad \lambda^{(k+1)} = \mathrm{Proj}_\Lambda \left( \lambda^{(k)} + \kappa_2 \left( \mathcal{E}_n(P_\phi) - \alpha_n \right) \right). \tag{4}$$

Here $\mathrm{Proj}_\Lambda$ is the projection to a pre-specified interval $\Lambda$ for $\lambda$, and $\mathcal{L}_{V,t}(\phi^{(k)}, \lambda^{(k)})$ is the Lagrangian function: $V_{P_{\phi^{(k)}}}^{\pi_t} + \lambda^{(k)}(\mathcal{E}_n(P_{\phi^{(k)}}) - \alpha_n)$. Here $\Lambda$ is a user-defined interval designed to prevent $\lambda^{(k)}$ from reaching 0 or $+\infty$.

## 4.2 Function approximation in MA

In the policy improvement step of algorithm 1, updating $\pi_{t+1}(\cdot \mid s)$ for an infinite number of states $s$ is computationally impossible, as $Q_{P_t}^{\pi_t}$ can only be evaluated when $\pi_t(\cdot \mid s)$ is known for all $s$. Although Monte Carlo estimation may be utilized in evaluating $Q_{P_t}^{\pi_t}$ in eq. (2), the computational complexity would grow exponentially with the number of iterations $T$, resulting in computational inefficiency. Notably, this issue is also present in Algorithm 1 of Xie et al. (2021). In contrast, our practical algorithm exhibits *polynomial* dependence on $T$, which is computationally efficient. A detailed justification for this claim is provided in Appendix 7.

By the definition of $D(\pi_t(\cdot \mid s), p)$, the objective function in the policy improvement step of algorithm 1 is equivalent to

$$\langle Q_{P_t}^{\pi_t}(s, \cdot), p \rangle + \frac{1}{\eta_t} \langle \nabla \omega \left( \pi_t(\cdot \mid s) \right), p \rangle - \frac{1}{\eta_t} \omega(p).$$

We define the *augmented* action-value function as

$$\tilde{\mathcal{Q}}_{\omega,t}(s, p) := \sum_{i=1}^m \tilde{Q}_{\omega,t}(s, A_i) p_i = \left\langle \tilde{Q}_{\omega,t}(s, \cdot), p \right\rangle,$$

where

$$\tilde{Q}_{\omega,t}(s, A_i) := Q_{P_t}^{\pi_t}(s, A_i) + \frac{1}{\eta_t} \nabla \omega(\pi_t(\cdot \mid s))_i$$

and $\tilde{Q}_{\omega,t}(s, \cdot)$ is a vector with its $i$-th element as $\tilde{Q}_{\omega,t}(s, A_i)$. The augmented action-value function not only incorporates information about the action-value function but also includes information about the policy itself. Following Lan (2022), we approximate $\tilde{\mathcal{Q}}_{\omega,t}(s, p)$ by a parametric function $f_t(s, p; \beta_t) \in \mathcal{F}_t$ such that $f_t(s, p; \beta_t^*) \approx \tilde{\mathcal{Q}}_{\omega,t}(s, p)$ for some $\beta_t^*$, which is sufficient to approximate $\tilde{Q}_{\omega,t}(s, A_i)$ for each $A_i$ according to $\tilde{\mathcal{Q}}_{\omega,t}(s, p) = \langle \tilde{Q}_{\omega,t}(s, \cdot), p \rangle$. To this end, for each $i = 1, ..., m$, we introduce $f_{t,i}(s; \beta_{t,i}) \in \mathcal{F}_{t,i}$ to approximate $\tilde{Q}_{\omega,t}(s, A_i)$, and thus

$$f_t(s, p; \beta_t) = \sum_{i=1}^m f_{t,i}(s; \beta_{t,i}) p_i := \langle f_t(s; \beta_t), p \rangle.$$

Here, $\mathcal{F}_{t,i}$ can be chosen as e.g. reproducing kernel Hilbert spaces (RKHS) or neural networks.

For each $i = 1, ..., m$, the optimal parameter $\beta_{t,i}^*$ can be obtained as follows,

$$\beta_{t,i}^* \in \mathrm{argmin}_{\beta_{t,i}} \mathbb{E}_{s \sim d_{P_t}^{\pi_t}} \left[ \left( \tilde{Q}_{\omega,t}(s, A_i) - f_{t,i}(s; \beta_{t,i}) \right)^2 \right]. \tag{5}$$

Specifically, we can generate $\{s_j\}_{j=1}^N \sim d_{P_t}^{\pi_t}$, and then minimize the empirical version of (5):

$$\widehat{\beta}_{t,i} \in \mathrm{argmin}_{\beta_{t,i}} \frac{1}{N} \sum_{j=1}^N \left[ \left( \tilde{Q}_{\omega,t}(s_j, A_i) - f_{t,i}(s_j; \beta_{t,i}) \right)^2 \right], \tag{6}$$

where $\{\tilde{Q}_{\omega,t}(s_j, A_i)\}_{j=1}^N$ are output from algorithm 4 (see Appendix E). The computable $\widehat{\beta}_{t,i}$ satisfies the property that $f_{t,i}(s; \widehat{\beta}_{t,i}) \approx f_{t,i}(s; \beta_{t,i}^*) \approx \tilde{Q}_{\omega,t}(s, A_i)$ for each $i = 1, ..., m$.

With the obtained $\{f_{t,i}(s;\widehat{\beta}_{t,i})\}_{i=1}^m$, the update rule in the policy improvement step can be written as below and solved by standard optimization algorithms,

$$\pi_{t+1}(\cdot \mid s) = \underset{p \in \Delta(\mathcal{A})}{\arg\max} \left\{ \sum_{i=1}^m f_{t,i}(s;\widehat{\beta}_{t,i})p_i - \frac{1}{\eta_t}\omega(p) \right\}, \forall s \in \mathcal{S}. \tag{7}$$

Such a design of function approximation enjoys several benefits. 1) The objective function in (7) is concave which can be solved by standard first-order optimization methods. 2) Compared to commonly-used parametric policy classes, our policy class is unrestricted, ensuring it includes the optimal policy. 3) The function approximation error in eq. (5) can be made arbitrarily small by enlarging the function classes $\mathcal{F}_{t,i}$.

### 4.3 An example

We provide a concrete example to illustrate the implementation procedure of the proposed practical algorithm 2, though our framework is general and different settings can be considered if desired.

**Policy evaluation step.** We consider the following empirical loss function $\widehat{L}_n$ for transition models:

$$\widehat{L}_n(P_\phi) = -\frac{1}{n}\sum_{i=1}^n \log P_\phi(s_i' \mid s_i, a_i).$$

Then $\widehat{P} = P_{\widehat{\phi}}$ is exactly the MLE, and

$$\mathcal{E}_n(P_\phi) = \frac{1}{n}\sum_{i=1}^n \log \frac{P_{\widehat{\phi}}(s_i' \mid s_i, a_i)}{P_\phi(s_i' \mid s_i, a_i)}.$$

The gradient of the Lagrangian function is:

$$\nabla_\phi \mathcal{L}_{V,t}(\phi, \lambda) = \nabla_\phi V_{P_\phi}^{\pi_t} + \lambda \nabla_\phi \mathcal{E}_n(P_\phi).$$

Here the model gradient $\nabla_\phi V_{P_\phi}^{\pi_t}$ can be calculated using the proposition 2 of Rigter et al. (2022). Specifically, let

$$M_\phi(s', s, a, \pi_t) := \left( r(s,a) + \gamma V_\phi^{\pi_t}(s') \right) \times \nabla_\phi \log P_\phi(s' \mid s, a),$$

then

$$\nabla_\phi V_{P_\phi}^{\pi_t} = \mathbb{E}_{s,a \sim d_{P_\phi}^{\pi_t}, s' \sim P_\phi(\cdot \mid s,a)} M_\phi(s', s, a, \pi_t).$$

For $\nabla_\phi \mathcal{E}_n(P_\phi)$, we have $\nabla_\phi \mathcal{E}_n(P_\phi) = -\frac{1}{n}\sum_{i=1}^n \nabla_\phi \log P_\phi(s_i' \mid s_i, a_i)$. Then the expressions of $\nabla_\phi \mathcal{L}_{V,t}(\phi, \lambda)$, $\nabla_\phi V_{P_\phi}^{\pi_t}$, and $\nabla_\phi \mathcal{E}_n(P_\phi)$ can be plugged into (4) and output a $P_t := P_{\phi^{(K)}}$.

**Policy improvement step.** We consider

$$\omega(p) := \sum_{i=1}^m p_i \log p_i$$

and introduce a multi-layer neural network $f_{t,i}(s;\beta_{t,i})$ for approximating $\tilde{Q}_{\omega,t}(s, A_i)$. For each $i = 1, ..., m$, in order to find the best $\beta_{t,i}$, we first sample $\{s_j\}_{j=1}^N$ i.i.d. from $d_{P_t}^{\pi_t}$, then run any policy evaluation procedure such as Monte Carlo (algorithm 4 in Appendix E) for $\tilde{Q}_{\omega,t}(s_j, A_i)$ for each $j = 1, ..., N$. Using training data $(s_j, \tilde{Q}_{\omega,t}(s_j, A_i))_{j=1}^N$, we can obtain $\widehat{\beta}_{t,i}$ by standard neural network (NN) training procedure. In addition, thanks to the form of $\omega$, we have a closed form solution to (7) which is the update rule

$$\pi_{t+1}(A_i \mid s) \propto \exp(\eta_t f_{t,i}(s;\widehat{\beta}_{t,i}))$$

for $i = 1, \ldots, m$. Besides NN for $f_{t,i}(s;\beta_{t,i})$ for each $i = 1, \ldots, m$, alternative general function classes, such as infinite-dimensional RKHS, can also be employed for function approximations.

### 4.4 Efficient policy update

We provide a detailed discussion of the computational complexity for each step in our MoMA algorithm here. We first consider the case $\omega(p) = \sum_{i=1}^{m} p_i \log p_i$, which leads to a closed function form of $\pi_{t+1}(A_i \mid s) \propto \exp\left(\eta_t f_{t,i}\left(s; \widehat{\beta}_{t,i}\right)\right)$ for each $i = 1, \ldots, m$. In the policy evaluation step $t$, given $\widehat{\beta}_{t-1}$ which is the output from the $(t-1)$-th iteration, we can count the number of calls of $\pi_t(s)$ from $t$ to $t+1$ as by realizing we need $\pi_t$ in the Monte Carlo evaluation of $V_{P_k}^{\pi_t}$ for $k = 1, \ldots, K$ and the sampling from $d_P^{\pi_t}$ in the policy evaluation step. Specifically, for each sampling or Monte Carlo evaluation, the effective numbers of using $\pi_t$ is $\frac{1}{1-\gamma}$, which is the effective trajectory length in an infinite-horizon discounted MDP. Therefore, for each $t$, in the policy evaluation step, we need to use $\pi_t$ for a total of $O\left(\frac{KL^2}{1-\gamma}\right)$ times, where $L$ denotes the number of Monte Carlo trajectories. In the policy improvement step, we need $\pi_t$ in the sampling of $(s_j, A_i)_{j=1}^{N}$ and Monte Carlo evaluation of $\tilde{Q}_{\omega,t}(s_j, A_i)$ for each $j, i$. Therefore, we need $O\left(\frac{NL}{(1-\gamma)^2}\right)$ Monte Carlo trajectories starting from $(s_j, p_j)$ for each $j = 1, \ldots, N$. Therefore, collectively at each $t$ in algorithm 2, we need to evaluate the function $\pi_t(A_i \mid \cdot) = \exp\left(f_{t-1,i}\left(\cdot; \widehat{\beta}_{t-1,i}\right)\right)/C$ approximately $O\left(\frac{KL^2}{1-\gamma} + \frac{NL}{(1-\gamma)^2}\right)$ times, which is independent of $t$. More generally, when $\omega$ does not induce an explicit solution to (7)), then $I$ more steps for gradient descent of (7) may be needed. In that case, running algorithm 2 costs $O\left(IT\left(\frac{KL^2}{1-\gamma} + \frac{NL}{(1-\gamma)^2}\right)\right)$ operations related to policy updates, which is polynomial on all the key parameters.

### 4.5 Extension to continuous action space

We now extend MoMA to handle complex RL tasks with nonlinear dynamics and continuous action spaces. Instead of considering $p \in \Delta(\mathcal{A})$ introduced in section 3.2 where $\mathcal{A}$ is assumed to be finite, we consider a continuous action space $\mathcal{A} \subset \mathbb{R}^{d_{\mathcal{A}}}$ in this section. Here $d_{\mathcal{A}}$ is the dimension of the action space and $\mathcal{A}$ is assumed to be a compact convex set. In the continuous-action case, we consider the deterministic policy $\pi : \mathcal{S} \to \mathcal{A}$, i.e. $\pi(s) \in \mathcal{A}$ is a feasible action for each state $s \in \mathcal{S}$.

In this case, a proposed update rule is

$$\pi_{t+1}(s) = \arg\max_{a \in \mathcal{A}} \left\{ Q_{P_t}^{\pi_t}(s, a) - \frac{1}{\eta_t} D\left(\pi_t(s), a\right) \right\} \quad \forall s. \tag{8}$$

Still, since the update rule (8) is computationally infeasible for infinitely many $s$, we propose a version with function approximation that is similar to section 4.2. Specifically, by expanding $D(\pi_t(s), a)$, the objective function in (8) is equivalent to $Q_{P_t}^{\pi_t}(s, a) + \frac{1}{\eta_t} \langle \nabla \omega(\pi_t(s)), a \rangle - \frac{1}{\eta_t} \omega(a)$. We also define $\tilde{\mathcal{Q}}_{\omega,t}(s, a) := Q_{P_t}^{\pi_t}(s, a) + \frac{1}{\eta_t} \langle \nabla \omega(\pi_t(s)), a \rangle$ as the *augmented action-value function*. We then approximate $\tilde{\mathcal{Q}}_{\omega,t}(s, a)$ by a parametric function $f_t(s, a; \beta_t) \in \mathcal{F}_t$ such that $f_t(s, a; \beta_t^*) \approx \tilde{\mathcal{Q}}_{\omega,t}(s, a)$ for some $\beta_t^*$. Here $\mathcal{F}_t$ can be RKHS or Neural Networks.

In particular, the optimal parameter $\beta_t^*$ can be obtained as follows,

$$\beta_t^* \in \arg\min_{\beta_t} \mathbb{E}_{(s,a) \sim d_{P_t}^{\pi_t}} \left[ \left( \tilde{Q}_{\omega,t}(s, a) - f_t(s, a; \beta_t) \right)^2 \right]. \tag{9}$$

Specifically, we can generate $\{s_j, a_j\}_{j=1}^{N} \sim d_{P_t}^{\pi_t}$, and then minimize the empirical version of (5):

$$\widehat{\beta}_t \in \arg\min_{\beta_t} \frac{1}{N} \sum_{j=1}^{N} \left[ \left( \tilde{Q}_{\omega,t}(s_j, a_j) - f_t(s_j, a_j; \beta_t) \right)^2 \right], \tag{10}$$

where $\{\tilde{Q}_{\omega,t}(s_j, a_j)\}_{j=1}^{N}$ are output from algorithm 5 (see Appendix E). The computable $\widehat{\beta}_t$ satisfies the property that $f_t(s, a; \widehat{\beta}_t) \approx f_t(s, a; \beta_t^*) \approx \tilde{Q}_{\omega,t}(s, a)$.

Finally, the update rule involving function approximation can be written as

$$\pi_{t+1}(s) = \arg\max_{a \in \mathcal{A}} \left\{ f_t(s, a; \widehat{\beta}_t) - \frac{1}{\eta_t} \omega(a) \right\}, \forall s \in \mathcal{S}. \tag{11}$$

A standard optimization procedure such as accelerated gradient descent method can be employed to solve (11).

For completeness, we summarize the whole algorithm for the continuous-action case in algorithm 3.

---

**Algorithm 3** MoMA: A Practical Algorithm in the continuous-action case
***
    **Input:** The learning rate $\eta_t$, $\mathcal{P}_{n,\alpha_n}$.
    **Initialization:** Initialize $\pi_0 = \mathrm{Unif}(\mathcal{A})$.
    **for** $t = 1$ **to** $T$ **do**
        Conservative policy evaluation:
        Compute $P_t := P_{\phi^{(K)}}$, where $\phi^{(K)}$ is the output from eq. (4).
        Policy improvement:
        Sample $\{s_j, a_j\}_{j=1}^N$ from $d_{P_t}^{\pi_t}$.
        **for** $j = 1$ **to** $N$ **do**
            Input $f_{t-1}(s, a; \widehat{\beta}_{t-1})$ and $(s_j, a_j)$ into algorithm 5, and output $\tilde{Q}_{\omega,t}(s_j, a_j)$.
        **end for**
        Find $\widehat{\beta}_{t,i}$ that solves eq. (10).
        Save the parametric function $f_t(s, a; \widehat{\beta}_t)$ as an input for iteration $t + 1$.
    **end for**

---

## 5 Theoretical analysis

In this section, we present the upper bound on the suboptimality of the learned policy $\widehat{\pi}$ in Algorithm 2 in terms of sample size, number of iterations and all key parameters. All proofs are presented in Appendix C. We first present the following assumptions.

**Assumption 1.** *The following conditions hold.*

*(a) (Data generation). The dataset $\mathcal{D} = (s_i, a_i, r_i, s_i')_{i=1}^n$ satisfies $(s_i, a_i) \overset{i.i.d.}{\sim} \rho$ with $s_i' \sim P^*(\cdot \mid s_i, a_i)$, where $\rho$ denotes the offline distribution induced by the behavior policy under $P^*$.*

*(b) (Coverage of any comparator policy $\pi^\dagger$). $C_{\pi^\dagger} := \sup_{s,a} \frac{d_{P^*}^{\pi^\dagger}(s,a)}{\rho(s,a)} < \infty$.*

*(c) (Realizability). $P^* \in \mathcal{P}$.*

*(d) There exist $\frac{c_1}{n} \leq \alpha_n = o(1)$, $\frac{c_2}{\sqrt{n}} \leq \delta_n = o(1)$ such that with high probabilities $P_* \in \mathcal{P}_{n,\alpha_n}$ and*

$$\varepsilon_{est} := \sup_{P \in \mathcal{P}_{n,\alpha_n}} \mathbb{E}_{(s,a)\sim\rho}[\|P(\cdot \mid s, a) - P^*(\cdot \mid s, a)\|_1] \leq \delta_n. \tag{12}$$

Assumption 1(a) is related to offline data generation, common in offline RL theoretical literature. Assumption 1(b) essentially requires the partial coverage of the offline distribution. The *concentrability coefficient* $C_{\pi^\dagger}$ measures the distribution mismatch between the offline distribution and the occupancy measure induced by $\pi^\dagger$. Assumption 1(c) requires that the model class $\mathcal{P}$ is sufficiently large such that there is no model misspecification error. Assumption 1(d) is needed to provide a fast statistical rate uniformly over the confidence set. We remark that assumption 1(d) is a mild condition. For example, commonly-used empirical risk functions (e.g. negative log-likelihood) satisfy it. See proposition 1, corollary 1 for more details in Appendix B.

Now we provide the suboptimality upper bound for algorithm 2 in the following theorem, assuming an access to a computational oracle for solving the contained minimization problem (1) in the conservative policy evaluation step. Theoretical results for algorithm 1, in which we assume no function approximation, are summarized in Theorem 2 in Appendix B.

**Theorem 1.** *Under Assumption 1, if $\eta_t = (1-\gamma)\sqrt{\frac{2\log(|\mathcal{A}|)}{T}}$ for every fixed $T$, then we have*

$$V_{P^*}^{\pi^\dagger} - V_{P^*}^{\widehat{\pi}} \lesssim \underbrace{\left(\frac{\gamma}{(1-\gamma)^2} + \frac{\gamma}{(1-\gamma)^3}\right)C_{\pi^\dagger}\ \varepsilon_{est}}_{\text{model error}} + \underbrace{\frac{1}{(1-\gamma)^2}\frac{1}{\sqrt{T}}}_{\text{policy optimization error}}$$

$$+ \underbrace{\frac{1}{(1-\gamma)^{\frac{3}{2}}}|\mathcal{A}|\sqrt{\sup_s\frac{d_{P^*}^{\pi^\dagger}(s)}{\mu_0(s)}}\left(\varepsilon_{approx} + \frac{\sqrt{\max_{t,i}|\mathcal{F}_{t,i}|}}{\sqrt{N}}\right)}_{\text{function approximation error}}$$

*with high probability. Here $\widehat{\pi} \sim \text{Unif}(\pi_0, \pi_1, ..., \pi_{T-1})$ where $\{\pi_t\}_{t=0}^{T-1}$ are output by Algorithm 2, and $\varepsilon_{approx}$ is defined in Definition A.3.*

A brief proof sketch is provided here. The proof of Theorem 1 is based on a decomposition of suboptimality:

$$V_{P^*}^{\pi^\dagger} - V_{P^*}^{\pi_t} = \underbrace{\left(V_{P^*}^{\pi^\dagger} - V_{P_t}^{\pi^\dagger}\right)}_{(a)} + \underbrace{\left(V_{P_t}^{\pi^\dagger} - V_{P_t}^{\pi_t}\right)}_{(b)} + \underbrace{\left(V_{P_t}^{\pi_t} - V_{P^*}^{\pi_t}\right)}_{(c)}. \tag{13}$$

Term (a) can be bounded by the distance between the true dynamic model $P^*$ and $P_t$, which converges to 0 as the sample size increases to $\infty$. Term (b) is managed through the mirror ascent update rule, which tends to 0 as the number of iterations $k \to \infty$. Term (c) is negative thanks to the construction of the confidence set and the definition of $P_t$. The proof is completed by averaging both sides of the above equation over $T$.

The upper bound in Theorem 1 includes three terms: a model error (depending on fixed $n$) coming from using offline data for estimation of the transition model, an optimization error (depending on iteration $T$) from the policy improvement, and a function approximation error coming from using Monte Carlo samples approximating the augmented $Q$ function. The model error is a finite-sample term that cannot be reduced under the offline setting, while the optimization error can be reduced when the number of iterations $T$ increases. Typically, we have $\varepsilon_{est} = O_P(1/\sqrt{n})$. The function approximation error involves an approximation error $\varepsilon_{approx}$ that decreases as the function class is enlarged $\max_{t,i}|\mathcal{F}_{t,i}| \to \infty$, an estimation error that scales with $O_P(1/\sqrt{N})$, and a distribution mismatch $\sup_s d_{P^*}^{\pi^\dagger}(s)/\mu_0(s)$ between the initial distribution and the occupancy measure induced by a single $\pi^\dagger$ under $P^*$. Indeed, if $\max_{t,i}|\mathcal{F}_{t,i}| \to \infty$, then $\varepsilon_{approx} \to 0$ by Definition A.3. Consequently, the function approximation error can converge to 0 as long as $\max_{t,i}|\mathcal{F}_{t,i}| \to \infty$ and $N \to \infty$ at the same speed based on its expression in Theorem 1. The function approximation error and the model error share similar intuitive interpretations. In the model error, $C_{\pi^\dagger}$ measures the transfer of $\varepsilon_{est}$ changing from the offline distribution to the target distribution $d_{P^*}^{\pi^\dagger}$. Analogously, $\sup_s d_{P^*}^{\pi^\dagger}(s)/\mu_0(s)$ in the function approximation error measures the transfer of $\varepsilon_{approx} + \frac{\sqrt{\max_{t,i}|\mathcal{F}_{t,i}|}}{\sqrt{N}}$ from the initial distribution to the target distribution $d_{P^*}^{\pi^\dagger}$.

## 6 Numerical studies

We perform numerical studies on both an illustrative synthetic dataset and MuJoCo (Todorov et al., 2012) benchmark datasets, where we extend MoMA to the continuous-action setting.

### 6.1 Synthetic dataset: an illustration

We design a test environment based on a modified random walk with terminal goal states to generate data with partial coverage and understand how pessimism helps MoMA avoid common pitfalls faced by model-based offline RL methods. $\omega(p)$ is set to be $\sum_{i=1}^{m} p_i \log(p_i)$.

**Environment and offline dataset** For each episode that starts with an initial state $s_0 \sim \mathcal{U}(-2, 2)$, at time $n$ a particle undergoes a random walk and transits according to a mixture of Gaussian dynamics: $s_{n+1} - s_n =: \Delta s \sim \psi_a \mathcal{N}(\mu_{1,a}, 0.1) + (1 - \psi_a)\mathcal{N}(\mu_{2,a}, 0.1)$, where the discrete action $a \in \{-1, 0, 1\}$ corresponds

to Left, Stay, and Right, respectively. We generate a partially covered offline dataset collected by a biased (to the left) behavioral policy $\beta$ that penalizes over-exploitation of the MLE. The full details for the environment and the behavioral policy are given in Appendix F.1.

**MoMA performance**   The implementation follows Algorithm 2, with details in Appendix F.1. We compare with 1) model-based NPG, which can be seen as a simplified version of MoMA without the conservative policy evaluation; 2) model-free neural fitted Q-iteration (NFQ) (Riedmiller, 2005); and 3) a uniformly random policy. All algorithms are offline trained to convergence, and then put into the environment for 1000 online evaluation episodes. We choose the number of episode steps as the metric for this shortest path problem, and report the means and standard deviations for the scores of all 4 algorithms in Table 1. MoMA has significantly superior performance compared to the baselines, while the model-based peer NPG achieves the second best performance. This is not surprising since a learned dynamics model in general helps generalization in tasks with continuous state spaces, and NFQ's lack of generalization ability is exacerbated in this partial coverage setting.

Table 1: Average episode length ($\pm$ std.) over 1000 online evaluation episodes; shorter is better.

| MoMA | NPG | NFQ | Uniform |
|---|---|---|---|
| **2.63 $\pm$ 1.61** | $3.20 \pm 2.33$ | $4.39 \pm 2.96$ | $6.13 \pm 5.07$ |

**Contribution from pessimism**   To understand pessimism empirically, we zoom into the state $s = 0.1$, where the optimal policy is consecutive Right, the data-supported suboptimal policy is consecutive Left, and the faulty policy centers on Stay. Due to inaccurate MLE, a model-based algorithm without pessimism over exploits the model and converges to the faulty action Stay. In contrast, pessimism allows MoMA to trust the model on Left which has high coverage, while cautiously modifying the model such that Stay does not lead to substantial Right movement, i.e., $\hat{\psi}_0$ increases (see Figure 2 in Appendix F). This behavior is clearly captured during the training process: shown in the right plot of Figure 1, while the weight of the faulty action Stay monotonically increases for NPG, it decreases from the 10$^{\text{th}}$ epoch for MoMA. As a result, the suboptimal action weight (shown middle) eventually dominates for MoMA but vanishes for NPG, and NPG's value function $V(0.1)$ under the true dynamics (shown left) decreases due to the over exploitation of the learned dynamics model.

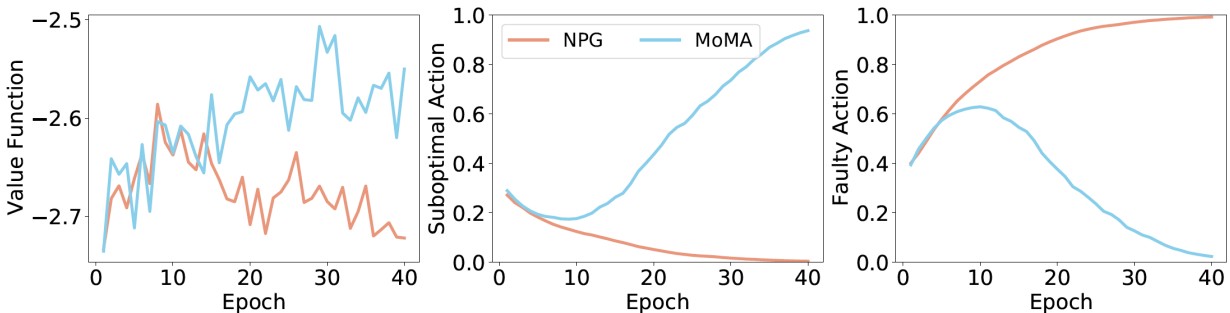

Figure 1: MoMA vs NPG training behavior, zoomed in at state 0.1. Left: the value function $V(0.1)$; middle: the data-supported suboptimal action weight; right: the model-mislead faulty action weight.

Furthermore, in Appendix F.2, we provide an illustrative example demonstrating that assuming a nonparametric policy class leads to better performance compared to a parametric class, highlighting scenarios where the limitations of parametric policies become evident.

## 6.2   Continuous action D4RL benchmark experiments

We now extend MoMA to handle complex RL tasks with nonlinear dynamics and continuous action spaces by approximating $\tilde{Q}_{\omega,t}(s,a)$, $\forall a \in$ continuous $\mathcal{A}$ rather than approximating $\tilde{Q}_{\omega,t}(s,A_i)$ for discrete actions.

See section 4.5 for more implementation details. We adjust the implmentation accordingly, and evaluate on D4RL (Fu et al., 2020) MuJoCo benchmark datasets.

We consider the medium, medium-replay, and medium-expert datasets for the Hopper, HalfCheetah, and Walker2D tasks (all v0), respectively. We compare against SOTA model-based baseline algorithms MOPO (Yu et al., 2020) and RAMBO (Rigter et al., 2022), and model-free baseline algorithms CQL (Kumar et al., 2020) and IQL (Kostrikov et al., 2021). With the exception of RAMBO for which we cite the results reported in Rigter et al. (2022), we train all algorithms for 1E6 steps with early stopping and with 5 different random seeds; additional experimental details are given in Appendix F.3. We summarize the scores (average returns of 10 evaluation episodes) in Table 2. MoMA consistently demonstrates performance that is at least comparable to state-of-the-art (SOTA) algorithms. Notably, in 4 out of 9 cases, our algorithm outperforms the other two model-based RL algorithms, achieving the highest performance.

Table 2: D4RL benchmark averaged performance over 5 random seeds ($\pm$ std.).

|  | Ours | Model-based | | Model-free | |
|---|---|---|---|---|---|
|  | **MoMA** | **MOPO** | **RAMBO** | **IQL** | **CQL** |
| Hopper, medium | $42.9 \pm 12.9$ | $58.2 \pm 15.2$ | $92.8 \pm 6.0$ | $101.1 \pm 0.5$ | $100.6 \pm 1.0$ |
| Hopper, medium-replay | $102.2 \pm 0.8$ | $101.2 \pm 0.9$ | $96.6 \pm 7.0$ | $66.0 \pm 16.2$ | $89.2 \pm 9.3$ |
| Hopper, medium-expert | $102.5 \pm 3.4$ | $46.3 \pm 17.1$ | $83.3 \pm 9.1$ | $106.7 \pm 7.1$ | $82.8 \pm 19.0$ |
| HalfCheetah,medium | $44.2 \pm 0.8$ | $26.5 \pm 10.1$ | $77.6 \pm 1.5$ | $42.5 \pm 0.1$ | $41.3 \pm 0.3$ |
| HalfCheetah, medium-replay | $55.4 \pm 1.0$ | $53.0 \pm 2.0$ | $68.9 \pm 2.3$ | $42.6 \pm 0.1$ | $45.9 \pm 0.1$ |
| HalfCheetah, medium-expert | $98.4 \pm 12.6$ | $95.5 \pm 10.6$ | $93.7 \pm 10.5$ | $96.9 \pm 1.8$ | $85.3 \pm 7.9$ |
| Walker2D medium | $36.8 \pm 20.4$ | $12.8 \pm 7.1$ | $86.9 \pm 2.7$ | $58.8 \pm 4.4$ | $83.1 \pm 0.8$ |
| Walker2D, medium-replay | $28.7 \pm 5.8$ | $67.9 \pm 5.8$ | $85.0 \pm 15.0$ | $22.5 \pm 11.1$ | $28.3 \pm 9.3$ |
| Walker2D, medium-expert | $98.5 \pm 5.5$ | $94.6 \pm 16.0$ | $68.3 \pm 20.6$ | $108.4 \pm 1.8$ | $106.3 \pm 15.5$ |

As one of the anonymous reviewers suggested, the confidence intervals (CIs) reported in our study can be biased due to being based on only a small number of seeds.

As a supplement to the standard mean of evaluation scores reported in Table 2, we have adopted the evaluation scheme proposed by Agarwal et al. (2021). Their method emphasizes the use of distributional metrics, which are less sensitive to outliers, providing a clearer and more robust picture of algorithm performance across different runs.

Specifically, we now include the interquartile mean (IQM) and the optimality gap as aggregate metrics, along with 95% bootstrap CIs. The IQM provides a robust measure of central tendency by focusing on the middle 50% of the data, reducing the influence of extreme values. The optimality gap measures how close the performance is to an optimal policy, offering a meaningful interpretation of results. Additionally, we present performance profiles based on score distributions, which offer a comprehensive view of the algorithm's performance across different scenarios. These results are detailed in Appendix F.2.

# 7 Comparisons with existing works

In this section, we present a discussion on the comparisons with some existing works in the field of offline reinforcement learning.

## 7.1 The policy evaluation step

In Section 3.1, we mentioned two fundamental advantages that can be attributed to the MoMA's design in the policy evaluation phase: 1) better expressiveness of the policy class, and 2) more flexibility of function approximations. We add more explanations about these two points here. First, we elaborate on better expressiveness of the policy class and more flexibility of the value function class. Indeed, these two advantages mainly stem from the construction of a confidence set as well as the separation of estimation and optimization under our model-based framework. Specifically, the offline dataset is only used to infer the transition model rather than directly infer the value function (which also depends on a policy). Thanks to this model-based

feature and the framework for the proposed algorithm, neither the size of the value function class nor that of the policy class is limited by the size of the offline dataset. In fact, the policy class in our settings can be taken large enough to contain the optimal policy, and the size of the value function class can keep growing until it contains the true value as long as we run Algorithm 4 enough times to generate sufficient Monte Carlo samples. These features result in an optimal rate of $O(1/\sqrt{n})$ as shown in Theorem 1, which outperforms the existing work (Xie et al., 2021). To illustrate this, we compare Corollary 5 of Section 4.1 in Xie et al. (2021) with Theorem 6.11 from our work, both discussing the suboptimality gaps under general function approximation. Corollary 5 in Xie et al. (2021) presented a convergence rate relative to the offline sample size $n$ as $O(1/n^{1/5})$, while Theorem 6.11 established a rate of $O(1/\sqrt{n})$. Therefore, MoMA enjoys the benefits of possessing more general policy classes and value function classes, while making no sacrifice on the data efficiency.

## 7.2 The policy improvement step

For the policy improvement step, we have developed the first computationally efficient algorithm under general function approximations (rather than linear approximations) with a theoretical guarantee. Existing literature is only computationally efficient either under linear approximation settings (Zanette et al., 2021; Xie et al., 2021), or without a theoretical guarantee for the policy improvement step (Cheng et al., 2022). We give detailed comparisons below.

Though Algorithm 1 of Xie et al. (2021) employs a mirror ascent method, it is not efficiently implementable when $|S| = \infty$, since it is impossible to enumerate every $s$ in a continuous state space to update the policy when the $f_t$ in Algorithm 1 of Xie et al. (2021) actually needs the access to $\pi_t(\cdot \mid s)$ for every $s$. Even if finitely many $\pi_t(s)$ are employed for obtaining $f_t$ via Monte Carlo methods, it still incurs an exponential complexity of at least $\Omega\left(C^T\right)$. To clearly show the difference, we exhibit our proposed algorithm and the one in Xie et al. (2021):

- Our proposed update rule when $\omega(p) = \sum_{i=1}^{m} p_i \log p_i$:

$$\pi_{t+1}\left(A_i \mid s\right) \propto \exp\left(\eta_t f_{t,i}(s; \widehat{\beta}_{t,i})\right)$$

  for each $i = 1, \ldots, m$.

- The update rule in Algorithm 1 in Xie et al. (2021):

$$\pi_{t+1}\left(A_i \mid s\right) \propto \exp\left(\eta_t \tilde{f}_t\left(s; \widehat{\beta}_{t,i}\right)\right) \pi_t\left(A_i \mid s\right)$$

  for each $i = 1, \ldots, m$.

In the update rule of Xie et al. (2021), $\pi_{t+1}\left(A_i \mid s\right)$ does not obtain a closed form without iteratively calling the previous iteration. In the continuous state space, this procedure is computationally inefficient. Moreover, assuming that the number of calls of $\pi_t(\cdot)$ used to approximate $\tilde{f}_t\left(\cdot; \widehat{\beta}_{t,i}\right)$ is $C$, then at least $\Omega\left(C^T\right)$ number of operations related to policies are needed. (See 4.4). In contrast, our algorithm's cost related to evaluating policies is $O\left(T\left(\frac{KL^2}{1-\gamma} + \frac{NL}{(1-\gamma)^2}\right)\right)$, evidently more computationally efficient.

Additionally, while Cheng et al. (2022) considers a parametric policy class, they do not provide a theoretical analysis for the actor step. Further, Zanette et al. (2021) only focuses on linear approximations, which may not be applicable in more general settings.

In summary, MoMA can be utilized when policy classes and value function classes of greater generality are needed, with no sacrifice on computational efficiency.

## 8 Conclusion

We developed MoMA, a model-based mirror ascent algorithm for offline RL, with general function approximation under the assumption of partial coverage. A key strength of MoMA is its ability to fully exploit

the potential of an unrestricted policy space, a characteristic advantage of model-based methods. This has been achieved through a combination of our theoretical framework and the application of mirror ascent techniques. Additionally, we have developed a practical, implementable version of MoMA that serves as an approximation of the theoretical algorithm. The efficacy of this practical algorithm has been demonstrated through a series of numerical experiments. While the numerical results do not consistently outperform the SOTA, one plausible reason is that the policy classes used in other offline algorithms might be sufficiently rich for these tasks. The proposed algorithm would gain more advantage in tasks where the optimal policy is harder to approximate. Improving the algorithm to consistently match the SOTA is an important area for future research.

There are several other intriguing avenues for future research. One promising area involves the development of algorithms that can theoretically address the computational challenges posed by the conservative evaluation step, particularly the need for a computational oracle. Another direction involves the integration of prior domain-specific knowledge, such as physical principles in mechanical systems or clinical insights in medical applications, into the transition model. Incorporating Bayesian estimators within our framework could provide a robust means of estimating transition models in these contexts. In addition, we can consider model misspecification, e.g., $P^* \notin \mathcal{P}$, analyze both the estimation error and approximation error, and investigate how they affect the suboptimality gap.

## Acknowledgements

We thank the Assigned Action Editor, Shixiang Gu, and the reviewers for their insightful comments and suggestions that significantly improve this paper. Yanxun Xu is supported in part by National Science Foundation grants 1918854 and 1940107, and National Institute of Health grant R01MH128085.

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

## A  Notations and definitions

**Definition A.1** (Integral Probability Metric (IPM)(Müller, 1997)). *$d_{\mathcal{F}}$ is an IPM defined by $\mathcal{F}$ if*

$$d_{\mathcal{F}}(P, Q) := \sup_{f \in \mathcal{F}} \left| \mathbb{E}_{x \sim P}\left[f\left(x\right)\right] - \mathbb{E}_{x \sim Q}\left[f\left(x\right)\right] \right|$$

*where $P$ and $Q$ are two probability measures.*

By considering different function class $\mathcal{F}$, we have the relationships between IPM and several popular measures of distance, such as TV and Wasserstein distance.

**Definition A.2** (Slater's condition). *The problem satisfies Slater's condition if it is strictly feasible, that is:*

$$\exists x_0 \in \mathcal{D} : f_i\left(x_0\right) < 0, \quad i = 1, \ldots, m, \quad h_i\left(x_0\right) = 0, \quad i = 1, \ldots, p$$

The next definition essentially measures the size of the function classes $\mathcal{F}_{t,i}$'s. If $\max_{t,i} |F_{t,i}|$ is sufficiently large, then $\varepsilon_{\text{approx}} \approx 0$.

**Definition A.3** (Approximation error).

$$\varepsilon_{\text{approx}} := \sup_{P, \pi, t, i} \inf_{f_{t,i} \in \mathcal{F}_{t,i}} \|\tilde{Q}_{\omega,t}(s, A_i) - f_{t,i}(s; \beta_{t,i})\|_{2, d_P^{\pi}}.$$

**Definition A.4** (Localized Population Rademacher Complexity). *(Wainwright, 2019, chap. 14). For a given radius $\delta > 0$ and function class $\mathcal{F}$, a localized population Rademacher complexity is defined as*

$$\bar{\mathcal{R}}_n(\delta; \mathcal{F}) = \mathbb{E}_{\varepsilon, x} \left[ \sup_{\substack{f \in \mathcal{F} \\ \|f\|_2 \leq \delta}} \left| \frac{1}{n} \sum_{i=1}^{n} \varepsilon_i f\left(x_i\right) \right| \right],$$

*where $\{x_i\}_{i=1}^{n}$ are i.i.d. samples from some underlying distribution $\mathbb{P}$, and $\{\varepsilon_i\}_{i=1}^{n}$ are i.i.d. Rademacher variables taking values in $\{-1, +1\}$ equiprobably, independent of the sequence $\{x_i\}_{i=1}^{n}$.*

**Definition A.5** (Star-shaped function class). *(Wainwright, 2019, chap. 14). A function class $\mathcal{F}$ is star-shaped around origin if for any $f \in \mathcal{F}$ and scalar $\alpha \in [0, 1]$, the function $\alpha f$ also belongs to $\mathcal{F}$.*

**Definition A.6** (Strongly convexity). *$f$ is strongly convex with modulus $\mu$ if the following holds:*

$$f(y) \geq f(x) + \nabla f(x)^T(y - x) + \frac{\mu}{2}\|y - x\|^2. \tag{14}$$

# B  Additional theoretical results

The following proposition shows that the commonly-used empirical risk functions satisfy Assumption 1(d).

**Proposition 1.** *Consider a uniformly bounded function class $\mathcal{L}(\mathcal{P}) := \{(s', s, a) \mapsto l(P, s', s, a), P \in \mathcal{P}\}$ that is star-shaped (defined in Appendix A) around the true $P^*$. Suppose $\delta_n^2 \geq \frac{c_1}{n}$ is a solution to the inequality $\bar{\mathcal{R}}_n(\delta; \mathcal{L}(\mathcal{P})) \leq \delta^2$ where $\bar{\mathcal{R}}_n(\delta; \mathcal{L}(\mathcal{P}))$ is the localized Rademacher complexity (defined in appendix A) of the function class. Assume $l(P, s', s, a)$ is $l_0$-Lipschitz w.r.t. $P$ , i.e.,*

$$l(P_1, s', s, a) - l(P_2, s', s, a) \leq l_0 |P_1(s'|s, a) - P_2(s'|s, a)|.$$

*Assume further that $l(P, s', s, a)$ is also strongly convex w.r.t. $P(s', s, a)$ under the norm $\|\cdot\|_{L^2, P^*}$. Suppose $H^2(P^*(\cdot \mid s, a), P(\cdot \mid s, a)) \leq c_3 \mathbb{E}_{s' \sim P^*(\cdot \mid s, a)} l(P, s', s, a) - c_3 \mathbb{E}_{s' \sim P^*} l(P^*, s', s, a)$. Let $\alpha = c_1 \delta_n^2$, then with high probabilities, we have $P^* \in \mathcal{P}_{n, \alpha_n}$ and*

$$\sup_{P \in \mathcal{P}_{n, \alpha_n}} \mathbb{E}_{(s, a) \sim \rho}[\|P(\cdot \mid s, a) - P^*(\cdot \mid s, a)\|_1] \leq c_2 \delta_n.$$

proposition 1 covers the commonly-used negative likelihood function classes, in which the empirical risk minimizers are exactly MLEs. We show such a construction satisfies assumption 1(d) in the following corollary.

**Corollary 1.** *Consider $l(P, s', s, a) := -\log P(s' \mid s, a)$ and $\widehat{L}_n(P) = -\frac{1}{n} \log P$. Assume there exist $b > 0, \nu > 0$ such that*

$$\sup_{P \in \mathcal{P}} \sup_{s', s, a} P(s'|s, a) < b, and, \inf_{s', s, a} P^*(s'|s, a) \geq \nu. \tag{15}$$

*If $\delta_n^2 \geq (1 + \frac{b}{\nu}) \frac{1}{n}$ solves the following inequality for the local Rademacher complexity of $\mathcal{P}$:*

$$\bar{\mathcal{R}}_n(\delta; \mathcal{P}) \leq \frac{\delta^2}{\sqrt{b + \nu}},$$

*then assumption 1(d) holds with $\alpha_n = c_1 \delta_n^2$ and $\epsilon_{est} = c_2 \delta_n$ for some constants $c_1, c_2$.*

In the following, we prove a suboptimality upper bound for the policy returned by algorithm 1.

**Theorem 2.** *Under assumption 1, if $\eta_t = (1 - \gamma) \sqrt{\frac{2 \log(|\mathcal{A}|)}{T}}$ for every fixed $T$, then we have*

$$V_{P^*}^{\pi^\dagger} - V_{P^*}^{\widehat{\pi}} \leq \underbrace{(\frac{\gamma}{(1 - \gamma)^2} + \frac{\gamma}{(1 - \gamma)^3}) C_{\pi^\dagger} \ \varepsilon_{est}}_{\text{statistical error}} + \underbrace{\frac{1}{(1 - \gamma)^2} \sqrt{\frac{2 \log(|\mathcal{A}|)}{T}}}_{\text{policy optimization error}}$$

*with high probability, where $\widehat{\pi} \sim \text{Unif}(\pi_0, \pi_1, ..., \pi_{T-1})$.*

The upper bound in Theorem 2 includes two terms: a statistical error (depending on fixed $n$) coming from using offline data for estimation, and an optimization error (depending on iteration $T$) coming from the policy improvement. Different from Theorem 1, function approximation is not involved in this case. The sacrifice is that the update rule is computationally infeasible.

# C  Technical proofs

In this section, we present all the technical proofs of the main theoretical results. We first prove theorem 2 in appendix C.1, which is a simplified version of theorem 1. Then we prove theorem 1 in appendix C.2 by further analyzing the effect of function approximation.

### C.1 Proofs of theorem 2

*Proof of theorem 2.* In this proof, we let $\Delta_m = \Delta(\mathcal{A})$ and $\pi(s) = \pi(\cdot \mid s)$ for clarity. We first split the average regret into the sum of three parts. We will deal with the three parts separately.

$$V_{P^*}^{\pi^\dagger} - \frac{1}{T}\sum_{t=0}^{T-1} V_{P^*}^{\pi_t} = \frac{1}{T}\sum_{t=0}^{T-1}\left(V_{P^*}^{\pi^\dagger} - V_{P_t}^{\pi^\dagger}\right) + \frac{1}{T}\sum_{t=0}^{T-1}\left(V_{P_t}^{\pi^\dagger} - V_{P_t}^{\pi_t}\right) + \frac{1}{T}\sum_{t=0}^{T-1}\left(V_{P_t}^{\pi_t} - V_{P^*}^{\pi_t}\right). \qquad (16)$$

For the first term, we can upper bound $V_{P^*}^{\pi^\dagger} - V_{P_t}^{\pi^\dagger}$ for each $t$. By the simulation lemma (1),

$$\begin{aligned}
V_{P^*}^{\pi^\dagger} - V_{P_t}^{\pi^\dagger} &= \frac{\gamma}{1-\gamma}\mathbb{E}_{(s,a)\sim d_{P^*}^{\pi^\dagger}}\left[\mathbb{E}_{s'\sim P^*(\cdot|s,a)}V_{P_t}^{\pi^\dagger}(s') - \mathbb{E}_{s'\sim P_t(\cdot|s,a)}V_{P_t}^{\pi^\dagger}(s')\right]\\
&\le \frac{\gamma}{1-\gamma}\mathbb{E}_{(s,a)\sim d_{P^*}^{\pi^\dagger}}\left[\left\|V_{P_t}^{\pi^\dagger}\right\|_\infty \|P^*(\cdot|s,a) - P_t(\cdot|s,a)\|_1\right]\\
&\le \frac{\gamma}{(1-\gamma)^2}\mathbb{E}_{(s,a)\sim d_{P^*}^{\pi^\dagger}}\|P^*(\cdot|s,a) - P_t(\cdot|s,a)\|_1\\
&\le \frac{\gamma}{(1-\gamma)^2}C_{\pi^\dagger}\mathbb{E}_{(s,a)\sim\rho}\|P^*(\cdot|s,a) - P_t(\cdot|s,a)\|_1\\
&\le \frac{\gamma}{(1-\gamma)^2}C_{\pi^\dagger}\varepsilon_{\text{est}}
\end{aligned} \qquad (17)$$

where we used the definitions of $C_{\pi^\dagger}$ and $\varepsilon_{\text{est}}$ (see assumption 1).

The third term in (16) is negative with high probability: by assumption 1(d), $P^* \in \mathcal{P}_{n,\alpha_n}$ with high probability. Recall the updating rule in (1), $P_t = \arg\min_{P\in\mathcal{P}_{n,\alpha_n}} V_P^{\pi_t}$. So $V_{P_t}^{\pi_t} \le V_{P^*}^{\pi_t}$ for all $t$ with high probability. Then the following holds with high probability:

$$\frac{1}{T}\sum_{t=0}^{T-1}\left(V_{P_t}^{\pi_t} - V_{P^*}^{\pi_t}\right) \le 0. \qquad (18)$$

Then it remains to upper bound $V_{P_t}^{\pi^\dagger} - V_{P_t}^{\pi_t}$. By performance difference lemma,

$$\begin{aligned}
\frac{1}{T}\sum_{t=0}^{T-1}\left(V_{P_t}^{\pi^\dagger} - V_{P_t}^{\pi_t}\right) &= \frac{1}{T(1-\gamma)}\sum_{t=0}^{T-1}\mathbb{E}_{(s,a)\sim d_{P_t}^{\pi^\dagger}}\left[A_{P_t}^{\pi_t}(s,a)\right]\\
&= \frac{1}{T(1-\gamma)}\sum_{t=0}^{T-1}\mathbb{E}_{(s,a)\sim d_{P^*}^{\pi^\dagger}}\left[A_{P_t}^{\pi_t}(s,a)\right]\\
&\quad + \frac{1}{T(1-\gamma)}\sum_{t=0}^{T-1}(\mathbb{E}_{(s,a)\sim d_{P_t}^{\pi^\dagger}} - \mathbb{E}_{(s,a)\sim d_{P^*}^{\pi^\dagger}})\left[A_{P_t}^{\pi_t}(s,a)\right].
\end{aligned} \qquad (19)$$

To further deal with the term above, we will first establish an upper bound for the advantage function $A_{P_t}^{\pi_t}(s,a)$.

Recall the policy update rule in (2),

$$\pi_{t+1}(s) = \arg\max_{p\in\Delta_m}\left\{\langle Q_{P_t}^{\pi_t}(s,\cdot), p\rangle - \frac{1}{\eta_t}D\left(\pi_t(s), p\right)\right\}$$

or equivalently,

$$\pi_{t+1}(s) = \arg\min_{p\in\Delta_m}\left\{-\langle Q_{P_t}^{\pi_t}(s,\cdot), p\rangle + \frac{1}{\eta_t}D\left(\pi_t(s), p\right)\right\}. \qquad (20)$$

By the optimality condition of (20), we have for any $p \in \Delta_m$,

$$\left\langle -Q_{P_t}^{\pi_t}(s,\cdot) + \frac{1}{\eta_t}\nabla_p D\left(\pi_t(s), p = \pi_{t+1}(s)\right), p - \pi_{t+1}(s)\right\rangle \ge 0.$$

Note that $D(\pi_t, p) = \omega(p) - \omega(\pi_t) - \langle \nabla \omega(\pi_t), p - \pi_t \rangle$. We can explicitly write out the gradient term in the inequality above, then we get

$$\left\langle -Q^{\pi_t}_{P_t}(s, \cdot) + \frac{1}{\eta_t} \Big( \nabla \omega(\pi_{t+1}(s)) - \nabla \omega(\pi_t(s)) \Big), p - \pi_{t+1}(s) \right\rangle \geq 0. \tag{21}$$

By definition of $D(\cdot, \cdot)$, we can derive that

$$D(\pi_t(s), p) - D(\pi_t(s), \pi_{t+1}(s)) - D(\pi_{t+1}(s), p) = \langle \nabla \omega(\pi_{t+1}(s)) - \nabla \omega(\pi_t(s)), p - \pi_{t+1}(s) \rangle.$$

So (21) becomes

$$\langle Q^{\pi_t}_{P_t}(s, \cdot), p - \pi_{t+1}(s) \rangle \leq \frac{1}{\eta_t} \Big( D(\pi_t(s), p) - D(\pi_t(s), \pi_{t+1}(s)) - D(\pi_{t+1}(s), p) \Big).$$

We can rewrite it in terms of advantage function:

$$\begin{aligned}
&\langle A^{\pi_t}_{P_t}(s, \cdot), p \rangle \\
&= \langle Q^{\pi_t}_{P_t}(s, \cdot), p \rangle - V^{\pi_t}_{P_t}(s) \\
&\leq \langle Q^{\pi_t}_{P_t}(s, \cdot), \pi_{t+1}(s) \rangle - V^{\pi_t}_{P_t}(s) + \frac{1}{\eta_t} \Big( D(\pi_t(s), p) - D(\pi_t(s), \pi_{t+1}(s)) - D(\pi_{t+1}(s), p) \Big) \\
&= \langle Q^{\pi_t}_{P_t}(s, \cdot), \pi_{t+1}(s) - \pi_t(s) \rangle + \frac{1}{\eta_t} \Big( D(\pi_t(s), p) - D(\pi_t(s), \pi_{t+1}(s)) - D(\pi_{t+1}(s), p) \Big).
\end{aligned}$$

Let $p = \pi^\dagger(s)$:

$$\begin{aligned}
&\langle A^{\pi_t}_{P_t}(s, \cdot), \pi^\dagger(s) \rangle \\
&\leq \langle Q^{\pi_t}_{P_t}(s, \cdot), \pi_{t+1}(s) - \pi_t(s) \rangle + \frac{1}{\eta_t} \Big( D(\pi_t(s), \pi^\dagger(s)) - D(\pi_t(s), \pi_{t+1}(s)) - D(\pi_{t+1}(s), \pi^\dagger(s)) \Big) \\
&\leq \langle Q^{\pi_t}_{P_t}(s, \cdot), \pi_{t+1}(s) - \pi_t(s) \rangle - \frac{1}{2\eta_t} \|\pi_{t+1}(s) - \pi_t(s)\|_1^2 \\
&\quad + \frac{1}{\eta_t} \Big( D(\pi_t(s), \pi^\dagger(s)) - D(\pi_{t+1}(s), \pi^\dagger(s)) \Big) \\
&\leq \|Q^{\pi_t}_{P_t}(s, \cdot)\|_\infty \|\pi_{t+1}(s) - \pi_t(s)\|_1 - \frac{1}{2\eta_t} \|\pi_{t+1}(s) - \pi_t(s)\|_1^2 \\
&\quad + \frac{1}{\eta_t} \Big( D(\pi_t(s), \pi^\dagger(s)) - D(\pi_{t+1}(s), \pi^\dagger(s)) \Big) \\
&\leq \frac{\eta_t}{2} \|Q^{\pi_t}_{P_t}(s, \cdot)\|_\infty^2 + \frac{1}{\eta_t} \Big( D(\pi_t(s), \pi^\dagger(s)) - D(\pi_{t+1}(s), \pi^\dagger(s)) \Big) \\
&\leq \frac{\eta_t}{2(1-\gamma)^2} + \frac{1}{\eta_t} \Big( D(\pi_t(s), \pi^\dagger(s)) - D(\pi_{t+1}(s), \pi^\dagger(s)) \Big)
\end{aligned} \tag{22}$$

where the third line above is because $D(p', p) \geq \frac{1}{2} \|p - p'\|^2$ (see section 3.2). For simplicity, we assume the norm is $L_1$-norm here. Even in the general case, recall that this norm $\|\cdot\|$ is defined on $\mathbb{R}^m$, and by a well known result in functional analysis, all norms on a finite dimension linear space are equivalent. So we can still establish a step similar to the third line in (22), replacing the second term by $-\frac{C}{2\eta_t} \|\pi_{t+1}(s) - \pi_t(s)\|_1^2$ for some constant $C$. Then in the last line in (22), the first term changes to $\frac{\eta_t}{2C} \|Q\|_\infty^2$ and the remaining are the same. We will see this difference does not affect the general form of the theorem, while it only changes some constant.

Then we can use (22) to upper bound the first term in (19):

$$
\begin{aligned}
&\sum_{t=0}^{T-1} \mathbb{E}_{(s,a)\sim d_{P^*}^{\pi^\dagger}} \left[ A_{P_t}^{\pi_t}(s,a) \right] \\
&= \sum_{t=0}^{T-1} \mathbb{E}_{s\sim d_{P^*}^{\pi^\dagger}} \left\langle A_{P_t}^{\pi_t}(s,\cdot), \pi^\dagger(s) \right\rangle \\
&\leq \sum_{t=0}^{T-1} \mathbb{E}_{s\sim d_{P^*}^{\pi^\dagger}} \left[ \frac{\eta_t}{2(1-\gamma)^2} + \frac{1}{\eta_t}\Big( D(\pi_t(s),\pi^\dagger(s)) - D(\pi_{t+1}(s),\pi^\dagger(s)) \Big) \right] \\
&= \frac{1}{2(1-\gamma)^2} \sum_{t=0}^{T-1} \eta_t + \mathbb{E}_{s\sim d_{P^*}^{\pi^\dagger}} \sum_{t=0}^{T-1} \Big( D(\pi_t(s),\pi^\dagger(s)) - D(\pi_{t+1}(s),\pi^\dagger(s)) \Big).
\end{aligned}
\tag{23}
$$

The second term in (23) can be bounded by the following telescoping technique. By assumption, $\{\eta_t\}$ is non-decreasing. Also note that the Bregman divergence is non-negative, so we have

$$
\begin{aligned}
&\sum_{t=0}^{T-1} \left( \frac{1}{\eta_t} D\left(\pi_t(s),\pi^\dagger(s)\right) - \frac{1}{\eta_t} D(\pi_{t+1}(s),\pi^\dagger(s)) \right) \\
&= \frac{1}{\eta_0} D\left(\pi_0(s),\pi^\dagger(s)\right) + \sum_{t=1}^{T-1} \left( \frac{1}{\eta_t} - \frac{1}{\eta_{t-1}} \right) D\left(\pi_t(s),\pi^\dagger(s)\right) - \frac{1}{\eta_{T-1}} D(\pi_T(s),\pi^\dagger(s)) \\
&\leq \frac{1}{\eta_0} D\left(\pi_0(s),\pi^\dagger(s)\right) \\
&\leq \frac{1}{\eta_0} D_0.
\end{aligned}
\tag{24}
$$

Then (23) becomes

$$
\sum_{t=0}^{T-1} \mathbb{E}_{(s,a)\sim d_{P^*}^{\pi^\dagger}} \left[ A_{P_t}^{\pi_t}(s,a) \right] \leq \frac{1}{2(1-\gamma)^2} \sum_{t=0}^{T-1} \eta_t + \frac{D_0}{\eta_0}.
\tag{25}
$$

The second term in (19) can be handled with simulation lemma: Let $\tilde{r}(s,a) = A_{P_t}^{\pi_t}(s,a)$. Consider two modified MDPs, $\widetilde{M}_t = (\mathcal{S}, \mathcal{A}, P_t, \tilde{r}, \gamma)$ and $\widetilde{M}^* = (\mathcal{S}, \mathcal{A}, P^*, \tilde{r}, \gamma)$. We still focus on the policy $\pi^\dagger$ and evaluate it under both modified MDPs. Since the visitation measure only depends on the transition probabilities and the discounting factor, we can rewrite the expectation of $\tilde{r}$ under visitation measure as the value function of modified MDP. Then directly apply simulation lemma:

$$
\begin{aligned}
&\frac{1}{1-\gamma} \left( \mathbb{E}_{(s,a)\sim d_{P_t}^{\pi^\dagger}} - \mathbb{E}_{(s,a)\sim d_{P^*}^{\pi^\dagger}} \right) A_{P_t}^{\pi_t}(s,a) \\
&= V_{\widetilde{M}_t}^{\pi^\dagger} - V_{\widetilde{M}^*}^{\pi^\dagger} \\
&= \frac{\gamma}{1-\gamma} \mathbb{E}_{(s,a)\sim d_{P^*}^{\pi^\dagger}} \left[ \mathbb{E}_{s'\sim P_t(\cdot|s,a)} V_{\widetilde{M}_t}^{\pi^\dagger}(s') - \mathbb{E}_{s'\sim P^*(\cdot|s,a)} V_{\widetilde{M}_t}^{\pi^\dagger}(s') \right].
\end{aligned}
$$

Note that the original reward function satisfies $r \in [0,1]$, so both $Q_{P_t}^{\pi_t}(\cdot, \cdot)$ and $V_{P_t}^{\pi_t}(\cdot)$ are bounded in $[0, \frac{1}{1-\gamma}]$. Then $|\tilde{r}| \leq \frac{1}{1-\gamma}$, $|V_{\tilde{M}_t}^{\pi^\dagger}(s')| \leq \frac{1}{(1-\gamma)^2}$. So

$$
\begin{aligned}
& \frac{1}{1-\gamma}\left(\mathbb{E}_{(s,a)\sim d_{P_t}^{\pi^\dagger}} - \mathbb{E}_{(s,a)\sim d_{P*}^{\pi^\dagger}}\right) A_{P_t}^{\pi_t}(s,a) \\
&= \frac{\gamma}{1-\gamma}\mathbb{E}_{(s,a)\sim d_{P*}^{\pi^\dagger}}\left[\mathbb{E}_{s'\sim P_t(\cdot|s,a)}V_{\tilde{M}_t}^{\pi^\dagger}(s') - \mathbb{E}_{s'\sim P^*(\cdot|s,a)}V_{\tilde{M}_t}^{\pi^\dagger}(s')\right] \\
&\leq \frac{\gamma}{1-\gamma}\mathbb{E}_{(s,a)\sim d_{P*}^{\pi^\dagger}}\left[\left\|V_{\tilde{M}_t}^{\pi^\dagger}\right\|_\infty \|P_t(\cdot|s,a) - P^*(\cdot|s,a)\|_1\right] \\
&\leq \frac{\gamma}{(1-\gamma)^3}\mathbb{E}_{(s,a)\sim d_{P*}^{\pi^\dagger}}\|P_t(\cdot|s,a) - P^*(\cdot|s,a)\|_1 \\
&\leq \frac{\gamma}{(1-\gamma)^3}C_{\pi^\dagger}\mathbb{E}_{(s,a)\sim\rho}\|P_t(\cdot|s,a) - P^*(\cdot|s,a)\|_1 \\
&\leq \frac{\gamma}{(1-\gamma)^3}C_{\pi^\dagger}\varepsilon_{\text{est}}
\end{aligned}
\tag{26}
$$

where the penultimate line comes from assumption 1(b), and the last step is by the definition of $\varepsilon_{\text{est}}$ (see assumption 1(d)).

So now we can use (25) and (26) to control the two terms in (19):

$$
\begin{aligned}
& \frac{1}{T}\sum_{t=0}^{T-1}\left(V_{P_t}^{\pi^\dagger} - V_{P_t}^{\pi_t}\right) \\
&= \frac{1}{T(1-\gamma)}\sum_{t=0}^{T-1}\mathbb{E}_{(s,a)\sim d_{P*}^{\pi^\dagger}}\left[A_{P_t}^{\pi_t}(s,a)\right] + \frac{1}{T(1-\gamma)}\sum_{t=0}^{T-1}(\mathbb{E}_{(s,a)\sim d_{P_t}^{\pi^\dagger}} - \mathbb{E}_{(s,a)\sim d_{P*}^{\pi^\dagger}})\left[A_{P_t}^{\pi_t}(s,a)\right] \\
&\leq \frac{1}{2T(1-\gamma)^3}\sum_{t=0}^{T-1}\eta_t + \frac{D_0}{T(1-\gamma)\eta_0} + \frac{\gamma}{(1-\gamma)^3}C_{\pi^\dagger}\varepsilon_{\text{est}}.
\end{aligned}
\tag{27}
$$

Finally, we use (17), (18), (27) to upper bound the three terms in (16):

$$
\begin{aligned}
& V_{P*}^{\pi^\dagger} - \frac{1}{T}\sum_{t=0}^{T-1}V_{P*}^{\pi_t} \\
&= \frac{1}{T}\sum_{t=0}^{T-1}\left(V_{P*}^{\pi^\dagger} - V_{P_t}^{\pi^\dagger}\right) + \frac{1}{T}\sum_{t=0}^{T-1}\left(V_{P_t}^{\pi^\dagger} - V_{P_t}^{\pi_t}\right) + \frac{1}{T}\sum_{t=0}^{T-1}\left(V_{P_t}^{\pi_t} - V_{P*}^{\pi_t}\right) \\
&\leq \frac{\gamma}{(1-\gamma)^2}C_{\pi^\dagger}\varepsilon_{\text{est}} + \frac{1}{2T(1-\gamma)^3}\sum_{t=0}^{T-1}\eta_t + \frac{D_0}{T(1-\gamma)\eta_0} + \frac{\gamma}{(1-\gamma)^3}C_{\pi^\dagger}\varepsilon_{\text{est}} + 0 \\
&\leq c_1 C_{\pi^\dagger}\varepsilon_{\text{est}} + \frac{1}{2T(1-\gamma)^3}\sum_{t=0}^{T-1}\eta_t + \frac{D_0}{T(1-\gamma)\eta_0}.
\end{aligned}
$$

$\square$

## C.2 Proofs for theorem 1

*Proof of theorem 1.* We first focus on $V_{P_t}^{\pi^\dagger} - V_{P_t}^{\pi_t}$. Let $\mathcal{A} = \{A_1, A_2, ..., A_m\}$, and the goal is to find an optimal randomized policy in the probability simplex $\Delta_m := \{p \in \mathbb{R}^m : \sum_{i=1}^m p_i = 1, p_i \geq 0, i = 1, \ldots, m\}$. Then, for a given $\pi_t(s) \in \Delta_m$, we use

$$
\begin{aligned}
Q_{P_t}^{\pi_t}(s, A_i) &:= R(s, A_i) + \gamma \int V_{P_t}^{\pi_t}(s') P(s' \mid s, A_i) \, ds', i = 1, \ldots, m, \\
\tilde{\mathcal{Q}}_{\omega,t}(s, A_i) &:= Q_{P_t}^{\pi_t}(s, A_i) + \frac{1}{\eta_t}\nabla_i \omega(\pi_t(s)), i = 1, \ldots, m
\end{aligned}
\tag{28}
$$

to denote the action value function and the augmented action value function evaluated at the action $A_i$. Then for any $p \in \Delta_m$

$$
\begin{aligned}
\tilde{\mathcal{Q}}_{\omega,t}(s,p) &= Q_{P_t}^{\pi_t}(s,p) + \langle \nabla \omega(\pi_t(s)), p \rangle / \eta_t \\
&= \sum_{i=1}^{m} Q_{P_t}^{\pi_t}(s,A_i) p_i + \langle \nabla \omega(\pi_t(s)), p \rangle / \eta_t \\
&:= \langle \tilde{\mathcal{Q}}_{\omega,t}(s,\cdot), p \rangle
\end{aligned}
\tag{29}
$$

where $\tilde{\mathcal{Q}}_{\omega,t}(s,\cdot)$ is defined as an $m$-dimensional vector with its $i$-th element as $\tilde{\mathcal{Q}}_{\omega,t}(s,A_i)$.

That means, if we want to approximate $\tilde{\mathcal{Q}}_{\omega,t}(s,p)$ which is a linear function of $\tilde{\mathcal{Q}}_{\omega,t}(s,\cdot)$, then we only need to approximate $\tilde{\mathcal{Q}}_{\omega,t}(s,A_i)$ for each $i = 1, ..., m$.

For each $i = 1, ..., m$, we consider the approximation

$$
f_{t,i}(s; \beta_{t,i}) \approx \tilde{\mathcal{Q}}_{\omega,t}(s, A_i)
\tag{30}
$$

where $f_{t,i}(s; \beta_{t,i})$ denotes some function class parameterized by $\beta$.

Then the update rule in (7) is reduced to

$$
\pi_{t+1}(s) = \arg\max_{p \in \Delta_m} \left\{ \langle f_t(s; \beta_t), p \rangle - \frac{1}{\eta_t} \omega(p) \right\}, \forall s \in \mathcal{S}.
\tag{31}
$$

For simplicity, we consider an equivalent rule of (31):

$$
\pi_{t+1}(s) = \arg\min_{p \in \Delta_m} \left\{ - \langle f_t(s; \beta_t), p \rangle + \frac{1}{\eta_t} \omega(p) \right\}, \forall s \in \mathcal{S}.
\tag{32}
$$

Here we notice that since $\omega$ is assumed to be strongly convex with modulus 1, and $\langle f_t(s; \beta), p \rangle$ is a convex function of $p$, we have a strongly convex objective function in (32) with modulus $\frac{1}{\eta_t}$. Also, $\Delta_m$ is a convex space. Therefore the optimization procedure in (32) is meaningful.

Then, by the optimality condition of (32), we have

$$
\left\langle \nabla \left\{ - \langle f_t(s; \beta_t), \pi_{t+1}(s) \rangle + \frac{1}{\eta_t} \omega(\pi_{t+1}(s)) \right\}, p - \pi_{t+1}(s) \right\rangle \geq 0
$$

for any $p \in \Delta_m$. By expanding this inequality, we get

$$
\begin{aligned}
\langle -f_t(s; \beta_t), p - \pi_{t+1}(s) \rangle &\geq -\frac{1}{\eta_t} \langle \nabla \omega(\pi_{t+1}(s)), p - \pi_{t+1}(s) \rangle \\
&= \frac{1}{\eta_t} \Big( D(\pi_{t+1}(s), p)) - \omega(p) + \omega(\pi_{t+1}(s)) \Big).
\end{aligned}
$$

We rewrite it as

$$
\begin{aligned}
&- \langle \tilde{\mathcal{Q}}_{\omega,t}(s,\cdot), p \rangle + \langle \tilde{\mathcal{Q}}_{\omega,t}(s,\cdot) - f_t(s; \beta_t), p \rangle \\
\geq &- \langle \tilde{\mathcal{Q}}_{\omega,t}(s,\cdot), \pi_{t+1}(s) \rangle + \langle \tilde{\mathcal{Q}}_{\omega,t}(s,\cdot) - f_t(s; \beta_t), \pi_{t+1}(s) \rangle \\
&+ \frac{1}{\eta_t} \omega(\pi_{t+1}(s)) - \frac{1}{\eta_t} \omega(p) + \frac{1}{\eta_t} D(\pi_{t+1}(s), p).
\end{aligned}
\tag{33}
$$

We notice that $\langle \tilde{\mathcal{Q}}_{\omega,t}(s,\cdot), p \rangle$ is exactly $\tilde{\mathcal{Q}}_{\omega,t}(s,p)$. For clarity, we denote

$$
\delta_t(s) := \tilde{\mathcal{Q}}_{\omega,t}(s,\cdot) - f_t(s; \beta_t)
$$

which represents the error in the approximation step in algorithm 2. By previous notations, $\delta_t(s)$ is also an $m$-dimension vector. Plugging $\langle \tilde{\mathcal{Q}}_{\omega,t}(s,\cdot), p \rangle = \tilde{\mathcal{Q}}_{\omega,t}(s,p) = Q_{P_t}^{\pi_t}(s,p) + \langle \nabla \omega(\pi_t(s)), p \rangle / \eta_t$ for any $p$, and

$\delta_t(s) = \tilde{\mathcal{Q}}_{\omega,t}(s,\cdot) - f_t(s;\beta_t)$ into (33), then we have

$$
\begin{aligned}
& - Q_{P_t}^{\pi_t}(s,p) - \langle \nabla \omega \left( \pi_t(s) \right), p \rangle / \eta_t + \langle \delta_t(s), p \rangle \\
\geq & - Q_{P_t}^{\pi_t}(s, \pi_{t+1}(s)) - \langle \nabla \omega \left( \pi_t(s) \right), \pi_{t+1}(s) \rangle / \eta_t + \langle \delta_t(s), \pi_{t+1}(s) \rangle \\
& + \frac{1}{\eta_t} \omega(\pi_{t+1}(s)) - \frac{1}{\eta_t} \omega(p) + \frac{1}{\eta_t} D(\pi_{t+1}(s), p).
\end{aligned}
\tag{34}
$$

By definition of $D(\cdot, \cdot)$, we have

$$
\omega \left( \pi_{t+1}(s) \right) - \omega(p) - \langle \nabla \omega \left( \pi_t(s) \right), \pi_{t+1}(s) - p \rangle = D \left( \pi_t(s), \pi_{t+1}(s) \right) - D \left( \pi_t(s), p \right).
\tag{35}
$$

Then (34) becomes

$$
\begin{aligned}
& Q_{P_t}^{\pi_t}(s, \pi_{t+1}(s)) - Q_{P_t}^{\pi_t}(s,p) \\
\geq & \langle \delta_t(s), \pi_{t+1}(s) - p \rangle + \frac{1}{\eta_t} D(\pi_{t+1}(s), p) + \frac{1}{\eta_t} D \left( \pi_t(s), \pi_{t+1}(s) \right) - \frac{1}{\eta_t} D \left( \pi_t(s), p \right).
\end{aligned}
\tag{36}
$$

Therefore we have

$$
\begin{aligned}
& A_{P_t}^{\pi_t}(s, \pi_{t+1}(s)) - A_{P_t}^{\pi_t}(s,p) \\
= & Q_{P_t}^{\pi_t}(s, \pi_{t+1}(s)) - V_{P_t}^{\pi_t}(s) - Q_{P_t}^{\pi_t}(s,p) + V_{P_t}^{\pi_t}(s) \\
= & Q_{P_t}^{\pi_t}(s, \pi_{t+1}(s)) - Q_{P_t}^{\pi_t}(s,p) \\
\geq & \langle \delta_t(s), \pi_{t+1}(s) - p \rangle + \frac{1}{\eta_t} D(\pi_{t+1}(s), p) + \frac{1}{\eta_t} D \left( \pi_t(s), \pi_{t+1}(s) \right) - \frac{1}{\eta_t} D \left( \pi_t(s), p \right).
\end{aligned}
\tag{37}
$$

Reorganize it and we have

$$
\begin{aligned}
& A_{P_t}^{\pi_t}(s,p) \\
\leq & A_{P_t}^{\pi_t}(s, \pi_{t+1}(s)) - \langle \delta_t(s), \pi_{t+1}(s) - p \rangle + \frac{1}{\eta_t} D \left( \pi_t(s), p \right) - \frac{1}{\eta_t} D(\pi_{t+1}(s), p) \\
& - \frac{1}{\eta_t} D \left( \pi_t(s), \pi_{t+1}(s) \right) \\
\leq & A_{P_t}^{\pi_t}(s, \pi_{t+1}(s)) - \langle \delta_t(s), \pi_{t+1}(s) - p \rangle + \frac{1}{\eta_t} D \left( \pi_t(s), p \right) - \frac{1}{\eta_t} D(\pi_{t+1}(s), p) \\
& - \frac{1}{2\eta_t} \| \pi_{t+1}(s) - \pi_t(s) \|_1^2 \\
= & \langle Q_{P_t}^{\pi_t}(s), \pi_{t+1}(s) - \pi_t(s) \rangle - \frac{1}{2\eta_t} \| \pi_{t+1}(s) - \pi_t(s) \|_1^2 \\
& - \langle \delta_t(s), \pi_{t+1}(s) - p \rangle + \frac{1}{\eta_t} D \left( \pi_t(s), p \right) - \frac{1}{\eta_t} D(\pi_{t+1}(s), p) \\
\leq & \| Q \|_\infty \| \pi_{t+1}(s) - \pi_t(s) \|_1 - \frac{1}{2\eta_t} \| \pi_{t+1}(s) - \pi_t(s) \|_1^2 \\
& - \langle \delta_t(s), \pi_{t+1}(s) - p \rangle + \frac{1}{\eta_t} D \left( \pi_t(s), p \right) - \frac{1}{\eta_t} D(\pi_{t+1}(s), p) \\
\leq & \frac{\eta_t}{2} \| Q \|_\infty^2 - \langle \delta_t(s), \pi_{t+1}(s) - p \rangle + \frac{1}{\eta_t} D \left( \pi_t(s), p \right) - \frac{1}{\eta_t} D(\pi_{t+1}(s), p)
\end{aligned}
\tag{38}
$$

where the third line above is because $D(p', p) \geq \frac{1}{2} \| p - p' \|^2$ (see section 3.2). The reason we assume $L_1$-norm was already explained in the proof of theorem 2 (see the remark after (22) in section appendix C.1).

Now we return to the analysis for $V_{P_t}^{\pi^\dagger} - V_{P_t}^{\pi_t}$:

$$\frac{1}{T+1}\sum_{t=0}^T (V_{P_t}^{\pi^\dagger} - V_{P_t}^{\pi_t})$$

$$= \frac{1}{T+1}\sum_{t=0}^T \frac{1}{1-\gamma}\mathbb{E}_{(s,a)\sim d_{P_t}^{\pi^\dagger}} A_{P_t}^{\pi_t}(s,a) \tag{39}$$

$$= \frac{1}{T+1}\sum_{t=0}^T \frac{1}{1-\gamma}\mathbb{E}_{(s,a)\sim d_{P*}^{\pi^\dagger}} A_{P_t}^{\pi_t}(s,a) + \frac{1}{T+1}\sum_{t=0}^T \frac{1}{1-\gamma}\Big(\mathbb{E}_{(s,a)\sim d_{P_t}^{\pi^\dagger}} - \mathbb{E}_{(s,a)\sim d_{P*}^{\pi^\dagger}}\Big) A_{P_t}^{\pi_t}(s,a).$$

For the first term in (39), we can let the randomized policy $p$ in (38) be $\pi^\dagger(s)$:

$$\frac{1}{T+1}\sum_{t=0}^T \frac{1}{1-\gamma}\mathbb{E}_{(s,a)\sim d_{P*}^{\pi^\dagger}} A_{P_t}^{\pi_t}(s,a)$$

$$= \frac{1}{T+1}\sum_{t=0}^T \frac{1}{1-\gamma}\mathbb{E}_{s\sim d_{P*}^{\pi^\dagger}} A_{P_t}^{\pi_t}(s,\pi^\dagger(s))$$

$$\leq \frac{1}{(T+1)(1-\gamma)}\sum_{t=0}^T \mathbb{E}_{s\sim d_{P*}^{\pi^\dagger}}\left[\frac{\eta_t}{2}\|Q\|_\infty^2 - \langle \delta_t(s), \pi_{t+1}(s) - \pi^\dagger(s)\rangle \right.$$

$$\left. + \frac{1}{\eta_t}D\big(\pi_t(s),\pi^\dagger(s)\big) - \frac{1}{\eta_t}D(\pi_{t+1}(s),\pi^\dagger(s))\right] \tag{40}$$

$$\leq \frac{1}{2(T+1)(1-\gamma)^3}\sum_{t=0}^T \eta_t + \frac{1}{(T+1)(1-\gamma)}\sum_{t=0}^T \mathbb{E}_{s\sim d_{P*}^{\pi^\dagger}} \langle \delta_t(s), \pi^\dagger(s) - \pi_{t+1}(s)\rangle$$

$$+ \frac{1}{(T+1)(1-\gamma)}\mathbb{E}_{s\sim d_{P*}^{\pi^\dagger}}\sum_{t=0}^T \left(\frac{1}{\eta_t}D\big(\pi_t(s),\pi^\dagger(s)\big) - \frac{1}{\eta_t}D(\pi_{t+1}(s),\pi^\dagger(s))\right).$$

The second term in (40) is bounded by approximation error:

$$\mathbb{E}_{s\sim d_{P*}^{\pi^\dagger}} \langle \delta_t(s), \pi^\dagger(s) - \pi_{t+1}(s)\rangle$$

$$= \mathbb{E}_{s\sim d_{P*}^{\pi^\dagger}} \langle \tilde{\mathcal{Q}}_{\omega,t}(s,\cdot) - f_t(s;\beta_t), \pi^\dagger(s) - \pi_{t+1}(s)\rangle$$

$$\leq \mathbb{E}_{s\sim d_{P*}^{\pi^\dagger}} \langle |\tilde{\mathcal{Q}}_{\omega,t}(s,\cdot) - f_t(s;\beta_t)|, \pi^\dagger(s)\rangle + \mathbb{E}_{s\sim d_{P*}^{\pi^\dagger}} \langle |\tilde{\mathcal{Q}}_{\omega,t}(s,\cdot) - f_t(s;\beta_t)|, \pi_{t+1}(s)\rangle$$

$$\leq 2|\mathcal{A}|\max_i \mathbb{E}_{s\sim d_{P*}^{\pi^\dagger}} \big|\tilde{\mathcal{Q}}_{\omega,t}(s,A_i) - f_{t,i}(s;\beta_{t,i})\big|$$

$$\leq 2|\mathcal{A}|\max_i \sqrt{\mathbb{E}_{s\sim d_{P*}^{\pi^\dagger}} \big(\tilde{\mathcal{Q}}_{\omega,t}(s,A_i) - f_{t,i}(s;\beta_{t,i})\big)^2}$$

$$\leq 2|\mathcal{A}|\max_i \left\|\frac{d_{P*}^{\pi^\dagger}(s)}{d_{P_t}^{\pi_t}(s)}\right\|_\infty^{\frac{1}{2}} \sqrt{\mathbb{E}_{s\sim d_{P_t}^{\pi_t}} \big(\tilde{\mathcal{Q}}_{\omega,t}(s,A_i) - f_{t,i}(s;\beta_{t,i})\big)^2}$$

$$\leq \frac{2|\mathcal{A}|\max_i}{\sqrt{1-\gamma}} \left\|\frac{d_{P*}^{\pi^\dagger}(s)}{\mu_0(s)}\right\|_\infty^{\frac{1}{2}} \sqrt{\mathbb{E}_{s\sim d_{P_t}^{\pi_t}} \big(\tilde{\mathcal{Q}}_{\omega,t}(s,A_i) - f_{t,i}(s;\beta_{t,i})\big)^2}.$$

By the updating rule in (6),

$$\mathbb{E}_{s \sim d_{P_t}^{\pi_t}} \left( \tilde{\mathcal{Q}}_{\omega,t}(s, A_i) - f_{t,i}(s; \beta_{t,i}) \right)^2$$

$$\leq \min_{\beta_{t,i}} \mathbb{E}_{s \sim d_{P_t}^{\pi_t}} \left( \tilde{\mathcal{Q}}_{\omega,t}(s, A_i) - f_{t,i}(s; \beta_{t,i}) \right)^2 + c \frac{|\mathcal{F}_{t,i}|}{N}, w.h.p.$$

$$\leq \sup_{P,\pi} \inf_{\beta_{t,i}} \mathbb{E}_{s \sim d_P^{\pi}} \left( \tilde{\mathcal{Q}}_{\omega,t}(s, A_i) - f_{t,i}(s; \beta_{t,i}) \right)^2 + c \frac{|\mathcal{F}_{t,i}|}{N}, w.h.p.$$

$$= \varepsilon_{\text{approx}}^2 + c \frac{|\mathcal{F}_{t,i}|}{N}, w.h.p.$$

where the second step follows from the analysis of standard M-estimator (Van de Geer & van de Geer, 2000) and the last step is because of definition A.3.

Combine the previous two inequalities, then we get

$$\mathbb{E}_{s \sim d_{P*}^{\pi\dagger}} \left\langle \delta_t(s), \pi^\dagger(s) - \pi_{t+1}(s) \right\rangle \leq \frac{2|\mathcal{A}|}{\sqrt{1-\gamma}} \left\| \frac{d_{P*}^{\pi\dagger}(s)}{\mu_0(s)} \right\|_\infty^{\frac{1}{2}} \left( \varepsilon_{\text{approx}} + c \sqrt{\frac{\max_{t,i} |\mathcal{F}_{t,i}|}{N}} \right). \tag{41}$$

The third term in (40) can be bounded by the same method from appendix C.1. (see (24)). So we have

$$\sum_{t=0}^{T} \left( \frac{1}{\eta_t} D \left( \pi_t(s), \pi^\dagger(s) \right) - \frac{1}{\eta_t} D(\pi_{t+1}(s), \pi^\dagger(s)) \right) \leq \frac{1}{\eta_0} D_0. \tag{42}$$

By (40), (41), (42), we obtain the following inequality, which is an upper bound for the first term in (39):

$$\frac{1}{T+1} \sum_{t=0}^{T} \frac{1}{1-\gamma} \mathbb{E}_{(s,a) \sim d_{P*}^{\pi\dagger}} A_{P_t}^{\pi_t}(s, a)$$

$$\leq \frac{1}{2(T+1)(1-\gamma)^3} \sum_{t=0}^{T} \eta_t + \frac{1}{(T+1)(1-\gamma)} \sum_{t=0}^{T} \mathbb{E}_{s \sim d_{P*}^{\pi\dagger}} \left\langle \delta_t(s), \pi^\dagger(s) - \pi_{t+1}(s) \right\rangle$$

$$+ \frac{1}{(T+1)(1-\gamma)} \mathbb{E}_{s \sim d_{P*}^{\pi\dagger}} \sum_{t=0}^{T} \left( \frac{1}{\eta_t} D \left( \pi_t(s), \pi^\dagger(s) \right) - \frac{1}{\eta_t} D(\pi_{t+1}(s), \pi^\dagger(s)) \right) \tag{43}$$

$$\leq \frac{1}{2(T+1)(1-\gamma)^3} \sum_{t=0}^{T} \eta_t + \frac{1}{1-\gamma} \frac{2|\mathcal{A}|}{\sqrt{1-\gamma}} \left\| \frac{d_{P*}^{\pi\dagger}(s)}{\mu_0(s)} \right\|_\infty^{\frac{1}{2}} \left( \varepsilon_{\text{approx}} + c \sqrt{\frac{\max_{t,i} |\mathcal{F}_{t,i}|}{N}} \right)$$

$$+ \frac{D_0}{(T+1)(1-\gamma)\eta_0}.$$

The second term in (39) can be handled by the same method from appendix C.1. (see (26)). So we have

$$\frac{1}{1-\gamma} \left( \mathbb{E}_{(s,a) \sim d_{P_t}^{\pi\dagger}} - \mathbb{E}_{(s,a) \sim d_{P*}^{\pi\dagger}} \right) A_{P_t}^{\pi_t}(s, a) \leq \frac{\gamma}{(1-\gamma)^3} C_{\pi\dagger} \varepsilon_{\text{est}}. \tag{44}$$

So now we can use (43) and (44) to control the two terms in (39) respectively:

$$\frac{1}{T} \sum_{t=0}^{T-1} (V_{P_t}^{\pi\dagger} - V_{P_t}^{\pi_t})$$

$$= \frac{1}{T} \sum_{t=0}^{T-1} \frac{1}{1-\gamma} \mathbb{E}_{(s,a) \sim d_{P*}^{\pi\dagger}} A_{P_t}^{\pi_t}(s, a) + \frac{1}{T} \sum_{t=0}^{T-1} \frac{1}{1-\gamma} \left( \mathbb{E}_{(s,a) \sim d_{P_t}^{\pi\dagger}} - \mathbb{E}_{(s,a) \sim d_{P*}^{\pi\dagger}} \right) A_{P_t}^{\pi_t}(s, a) \tag{45}$$

$$\leq \frac{1}{2T(1-\gamma)^3} \sum_{t=0}^{T-1} \eta_t + \frac{1}{1-\gamma} \frac{2|\mathcal{A}|}{\sqrt{1-\gamma}} \left\| \frac{d_{P*}^{\pi\dagger}(s)}{\mu_0(s)} \right\|_\infty^{\frac{1}{2}} \varepsilon_{\text{approx}} + \frac{D_0}{T(1-\gamma)\eta_0} + \frac{\gamma}{(1-\gamma)^3} C_{\pi\dagger} \varepsilon_{\text{est}}.$$

Finally, we consider $V_{P^*}^{\pi^\dagger} - V_{P^*}^{\pi_t}$:

$$V_{P^*}^{\pi^\dagger} - \frac{1}{T} \sum_{t=0}^{T-1} V_{P^*}^{\pi_t} = \frac{1}{T} \sum_{t=0}^{T-1} \left( V_{P^*}^{\pi^\dagger} - V_{P_t}^{\pi^\dagger} \right) + \frac{1}{T} \sum_{t=0}^{T-1} \left( V_{P_t}^{\pi^\dagger} - V_{P_t}^{\pi_t} \right) + \frac{1}{T} \sum_{t=0}^{T-1} \left( V_{P_t}^{\pi_t} - V_{P^*}^{\pi_t} \right). \tag{46}$$

The second term in (46) is already upper bounded by (45).

The third term in (46) is negative with high probability: by assumption 1(c), $P^* \in \mathcal{P}_{n,\alpha_n}$ with high probability. Recall the updating rule in (1), $P_t = \operatorname{argmin}_{P \in \mathcal{P}_{n,\alpha_n}} V_P^{\pi_t}$. So $V_{P_t}^{\pi_t} \le V_{P^*}^{\pi_t}$ for all $t$ with high probability. Then the following holds with high probability:

$$\frac{1}{T} \sum_{t=0}^{T-1} \left( V_{P_t}^{\pi_t} - V_{P^*}^{\pi_t} \right) \le 0. \tag{47}$$

The first term in (46) can be dealt by simulation lemma, which is same to (17) in appendix C.1:

$$V_{P^*}^{\pi^\dagger} - V_{P_t}^{\pi^\dagger} \le \frac{\gamma}{(1-\gamma)^2} C_{\pi^\dagger} \varepsilon_{\text{est}}. \tag{48}$$

By (45), (46), (47), (48),

$$\begin{aligned}
V_{P^*}^{\pi^\dagger} - \frac{1}{T} \sum_{t=0}^{T-1} V_{P^*}^{\pi_t} \le{}& \frac{\gamma}{(1-\gamma)^2} C_{\pi^\dagger} \varepsilon_{\text{est}} + \frac{1}{2T(1-\gamma)^3} \sum_{t=0}^{T-1} \eta_t \\
&+ \frac{1}{1-\gamma} \frac{2|\mathcal{A}|}{\sqrt{1-\gamma}} \left\| \frac{d_{P^*}^{\pi^\dagger}(s)}{\mu_0(s)} \right\|_\infty^{\frac{1}{2}} \left( \varepsilon_{\text{approx}} + c\sqrt{\frac{\max_{t,i} |\mathcal{F}_{t,i}|}{N}} \right) \\
&+ \frac{D_0}{T(1-\gamma)\eta_0} + \frac{\gamma}{(1-\gamma)^3} C_{\pi^\dagger} \varepsilon_{\text{est}} + 0 \\
\le{}& c_1 C_{\pi^\dagger} \varepsilon_{\text{est}} + c_2 |\mathcal{A}| \sqrt{\sup_s \frac{d_{P^*}^{\pi^\dagger}(s)}{\mu_0(s)}} \left( \varepsilon_{\text{approx}} + c\sqrt{\frac{\max_{t,i} |\mathcal{F}_{t,i}|}{N}} \right) \\
&+ \frac{1}{2T(1-\gamma)^3} \sum_{t=0}^{T-1} \eta_t + \frac{D_0}{T(1-\gamma)\eta_0}.
\end{aligned}$$

$\square$

## C.3   Proofs of proposition 1

*Proof.* Here we adapt a result from theorem 3. Specifically, given the conditions in proposition 1, we have

$$\|\widehat{P} - P^*\|_2 \le c_1 \delta_n$$

and

$$\mathbb{E}(l(\hat{P}) - l(P^*)) \le c_2 \delta_n^2.$$

Now we prove the first result in proposition 1, i.e., $P^* \in \mathcal{P}_{n,\alpha_n}$ with high probability. Consider $\widehat{L}_n(P^*) - \widehat{L}_n(\widehat{P})$ where $\widehat{P}$ minimize $\widehat{L}_n(P)$, and $\widehat{L}_n(P) = \frac{1}{n} \sum_{i=1}^n l(P)(s_i, a_i, s_i')$. We also use the notion $L(P) := \mathbb{E}l(P)$ which is an population counterpart of $\widehat{L}_n$. Then we have

$$\begin{aligned}
& \widehat{L}_n(P^*) - \widehat{L}_n(\widehat{P}) \\
={}& \widehat{L}_n(P^*) - L(P^*) + L(P^*) - L(\widehat{P}) + L(\widehat{P}) - \widehat{L}_n(\widehat{P}) \\
={}& (\widehat{L}_n(P^*) - \widehat{L}_n(\widehat{P}) + L(\widehat{P}) - L(P^*)) + L(P^*) - L(\widehat{P}) \\
\le{}& |(\widehat{L}_n(P^*) - \widehat{L}_n(\widehat{P}) + L(\widehat{P}) - L(P^*))|.
\end{aligned} \tag{49}$$

as the third term is less than 0.

By (b) of theorem 3, we have

$$\frac{|(\widehat{L}_n(P^*) - \widehat{L}_n(\widehat{P}) + L(\widehat{P}) - L(P^*))|}{\|\widehat{P} - P^*\|_2} \leq \sup_P \frac{|(\widehat{L}_n(P^*) - \widehat{L}_n(P) + L(P) - L(P^*))|}{\|P - P^*\|_2} \leq c_3 \delta_n.$$

Then we get

$$|(\widehat{L}_n(P^*) - \widehat{L}_n(\widehat{P}) + L(\widehat{P}) - L(P^*))| \leq c_3 \delta_n \|\widehat{P} - P^*\|_2 \leq c_4 \delta_n^2.$$

Therefore we get $\widehat{L}_n(P^*) - \widehat{L}_n(\widehat{P}) \leq c_4 \delta_n^2$, and it implies

$$P^* \in \mathcal{P}_{n,\alpha_n}$$

where $\alpha$ is set to be $c\delta_n^2$.

Next, we show that $\mathbb{E}_{s,a\sim\rho}\|P(\cdot|s,a) - P^*(\cdot|s,a)\|_1 \leq c\delta_n$ for every $P \in \mathcal{P}_{n,\alpha_n}$. Here we incorporate an assumption that $\mathbb{E}_{s,a\sim\rho}H^2(P,P^*) \leq L(P) - L(P^*)$ for every $P$. Then we have

$$
\begin{aligned}
&\mathbb{E}_{s,a\sim\rho}H^2(P,P^*) \\
&\leq L(P) - L(P^*) \\
&= L(P) - \widehat{L}_n(P) + \widehat{L}_n(P) - \widehat{L}_n(P^*) + \widehat{L}_n(P^*) - L(P^*) \\
&\leq |L(P) - \widehat{L}_n(P) + \widehat{L}_n(P^*) - L(P^*)| + |\widehat{L}_n(P) - \widehat{L}_n(P^*)| \\
&\leq |L(P) - \widehat{L}_n(P) + \widehat{L}_n(P^*) - L(P^*)| + |\widehat{L}_n(P) - \widehat{L}_n(\widehat{P})| + |\widehat{L}_n(\widehat{P}) - \widehat{L}_n(P^*)| \\
&\leq c_1\delta_n^2 + c_2\alpha_n + c_2\alpha_n \\
&\leq c\delta_n^2.
\end{aligned}
\tag{50}
$$

Since $H^2$ is an upper bound for TV distance, we have

$$\sup_{P\in\mathcal{P}_{n,\alpha_n}} \mathbb{E}_{s,a\sim\rho}\|P(\cdot|s,a) - P^*(\cdot|s,a)\|_1 \leq c\delta_n.$$

And the proof is done. □

### C.4 Proofs of corollary 1

*Proof.* Let

$$\widehat{L}_n(P) = \frac{1}{n}\sum_{i=1}^n \frac{P^*(s_i' \mid s_i, a_i)}{P(s_i' \mid s_i, a_i)}$$

and its population counterpart:

$$L(P) = \mathbb{E}_{(s,a)\sim\rho, s'\sim P^*(\cdot|s,a)} \frac{P^*(s' \mid s, a)}{P(s' \mid s, a)} = D_{\mathrm{KL}}(P^*\|P).$$

We prove $P^* \in \mathcal{P}_{n,\alpha_n}$ first. Consider $\widehat{L}_n(P^*) - \widehat{L}_n(\widehat{P})$ where $\widehat{P}$ minimize $\widehat{L}_n(P)$. Then we have

$$
\begin{aligned}
&\widehat{L}_n(P^*) - \widehat{L}_n(\widehat{P}) \\
&= \widehat{L}_n(P^*) - L(P^*) + L(P^*) - L(\widehat{P}) + L(\widehat{P}) - \widehat{L}_n(\widehat{P}) \\
&= (a) + (b) + (c)
\end{aligned}
\tag{51}
$$

Terms (a)(c) can be bounded by

$$\sup_{P\in\mathcal{P}} |(\widehat{L}_n - L)(P)|.$$

Again, we use theorem 3 to show that $\sup_{P\in\mathcal{P}} |(\widehat{L}_n - L)(P)| \leq \delta_n^2$.

For term $(b)$, we notice that $L(P^*) - L(\widehat{P}) = D_{\mathrm{KL}}(P^*\|\widehat{P})$. Combining lemma 3 which shows convergence rate of MLE under Hellinger distance and lemma 4, which upper bounds KL divergence by Hellinger distance when $P$ has a lower bound, then we have

$$(b) = D_{\mathrm{KL}}(P^*\|\widehat{P}) \le \delta_n^2$$

with high probability.

Then we have shown that

$$\widehat{L}_n(P^*) - \widehat{L}_n(\widehat{P}) \le c\delta_n^2 = \alpha$$

with probability at least $1 - \delta$. This implies that $P^* \in \mathcal{P}_\alpha$ with probability at least $1 - \delta$.

For the second part, we show

$$\sup_{P \in \mathcal{P}_{n,\alpha_n}} (\mathbb{E}_{(s,a)\sim\rho}[d_f(P(\cdot \mid s,a), P^*(\cdot \mid s,a))^2])^{\frac{1}{2}} \le c_2\delta_n. \tag{52}$$

To see this, we bound the Hellinger distance by KL divergence (lemma 4), specifically, for any $P \in \mathcal{P}_{n,\alpha_n}$ we have

$$
\begin{aligned}
H^2(P, P^*) &\le KL(P\|P^*) \\
&= L(P) - L(P^*) \\
&= L(P) - \widehat{L}_n(P) + \widehat{L}_n(P) - \widehat{L}_n(\widehat{P}) + \widehat{L}_n(\widehat{P}) - L(\widehat{P}) + L(\widehat{P}) - L(P^*).
\end{aligned}
\tag{53}
$$

Again, the first and the third terms are bounded by

$$\sup_{P \in \mathcal{P}} |(\widehat{L}_n - L)(P)| \le \delta_n^2.$$

The second term is bounded by $\alpha = c\delta_n^2$ because $P \in \mathcal{P}_{n,\alpha_n}$. The fourth term is equal to $KL(\widehat{P}\|P^*) \le c_3\delta_n^2$ by consistency of MLE in KL-divergence. And the proof is done. $\qquad\square$

## D  Supporting lemmas

**Lemma 1** (A generalization of simulation lemma). *Suppose $\mathcal{S}$, $\mathcal{A}$, $r$, $\gamma$, $\mu_0$ are all fixed. Here $\mathcal{S}$ and $\mathcal{A}$ can be infinite sets, and $r : \mathcal{S} \to \mathbb{R}$ can be any real value function. For two arbitrary transition models $P$ and $\widehat{P}$, and any policy $\pi : \mathcal{S} \to \Delta(\mathcal{A})$, we have*

$$V_P^\pi - V_{\widehat{P}}^\pi = \frac{\gamma}{1-\gamma}\mathbb{E}_{(s,a)\sim d_P^\pi}\left[\mathbb{E}_{s'\sim P(\cdot|s,a)}\left[V_{\widehat{P}}^\pi(s')\right] - \mathbb{E}_{s'\sim\widehat{P}(\cdot|s,a)}\left[V_{\widehat{P}}^\pi(s')\right]\right].$$

*If $V_{\widehat{P}}^\pi(s)$ is bounded, i.e. $-C \le V_{\widehat{P}}^\pi(s) \le C$, $\forall s \in \mathcal{S}$, then we further have*

$$\left|V_P^\pi - V_{\widehat{P}}^\pi\right| \le 2C\frac{\gamma}{1-\gamma}\mathbb{E}_{(s,a)\sim d_P^\pi}\left[\mathrm{TV}(P(\cdot|s,a),\widehat{P}(\cdot|s,a))\right].$$

*If $V_{\widehat{P}}^\pi(s)$ is positive and bounded, i.e. $0 \le V_{\widehat{P}}^\pi(s) \le C$, $\forall s \in \mathcal{S}$, then*

$$\left|V_P^\pi - V_{\widehat{P}}^\pi\right| \le C\frac{\gamma}{1-\gamma}\mathbb{E}_{(s,a)\sim d_P^\pi}\left[\mathrm{TV}(P(\cdot|s,a),\widehat{P}(\cdot|s,a))\right].$$

*Proof.* We first prove the first part of the lemma.

Let $d_P^\pi(\cdot,\cdot|s_0,a_0)$ denote the visitation measure over $(s,a)$ conditioning on $(S_0 = s_0, A_0 = a_0)$ under transition model $P$, i.e. $d_P^\pi(\cdot,\cdot|s_0,a_0) = (1-\gamma)\sum_{t=0}^\infty \gamma^t P^\pi(S_t = \cdot\,, A_t = \cdot\mid s_0,a_0)$.

Then we have for any $(s_0,a_0)$,

$$Q_P^\pi(s_0,a_0) = \frac{1}{1-\gamma}\mathbb{E}_{(s,a)\sim d_P^\pi(\cdot,\cdot|s_0,a_0)}[r(s,a)]. \tag{54}$$

By Bellman equation, for any $(s, a)$,

$$Q_P^\pi(s, a) = r(s, a) + \gamma \mathbb{E}_{s' \sim P(\cdot|s,a), a' \sim \pi(\cdot|s')} \left[ Q_P^\pi(s', a') \right]. \tag{55}$$

$$Q_{\widehat{P}}^\pi(s, a) = r(s, a) + \gamma \mathbb{E}_{s' \sim \widehat{P}(\cdot|s,a), a' \sim \pi(\cdot|s')} \left[ Q_{\widehat{P}}^\pi(s', a') \right]. \tag{56}$$

Substitute the $r(s, a)$ in (54) by the $r(s, a)$ in (56):

$$Q_P^\pi(s_0, a_0) = \frac{1}{1 - \gamma} \mathbb{E}_{(s,a) \sim d_P^\pi(\cdot,\cdot|s_0,a_0)} \left[ Q_{\widehat{P}}^\pi(s, a) - \gamma \mathbb{E}_{s' \sim \widehat{P}(\cdot|s,a), a' \sim \pi(\cdot|s')} Q_{\widehat{P}}^\pi(s', a') \right]. \tag{57}$$

By (54) and (55), we first apply (55) to the $Q_P^\pi(s_0, a_0)$ in (54), then apply (54) iteratively:

$$\frac{1}{1 - \gamma} \mathbb{E}_{(s,a) \sim d_P^\pi(\cdot,\cdot|s_0,a_0)}[r(s, a)]$$
$$= Q_P^\pi(s_0, a_0)$$
$$= r(s_0, a_0) + \gamma \mathbb{E}_{s \sim P(\cdot|s_0,a_0), a \sim \pi(\cdot|s)} \left[ Q_P^\pi(s, a) \right]$$
$$= r(s_0, a_0) + \gamma \mathbb{E}_{s \sim P(\cdot|s_0,a_0), a \sim \pi(\cdot|s)} \left[ \frac{1}{1 - \gamma} \mathbb{E}_{(s',a') \sim d_P^\pi(\cdot,\cdot|s,a)}[r(s', a')] \right].$$

Rearrange it as

$$-r(s_0, a_0) = \frac{\gamma}{1 - \gamma} \mathbb{E}_{s \sim P(\cdot|s_0,a_0), a \sim \pi(\cdot|s)} \left[ \mathbb{E}_{(s',a') \sim d_P^\pi(\cdot,\cdot|s,a)}[r(s', a')] \right]$$
$$- \frac{1}{1 - \gamma} \mathbb{E}_{(s,a) \sim d_P^\pi(\cdot,\cdot|s_0,a_0)}[r(s, a)].$$

Note that the equation above holds for any real function $r : \mathcal{S} \times \mathcal{A} \to \mathbb{R}$, so we can replace $r(\cdot, \cdot)$ by $Q_{\widehat{P}}^\pi(\cdot, \cdot)$

$$-Q_{\widehat{P}}^\pi(s_0, a_0) = \frac{\gamma}{1 - \gamma} \mathbb{E}_{s \sim P(\cdot|s_0,a_0), a \sim \pi(\cdot|s)} \left[ \mathbb{E}_{(s',a') \sim d_P^\pi(\cdot,\cdot|s,a)}[Q_{\widehat{P}}^\pi(s', a')] \right]$$
$$- \frac{1}{1 - \gamma} \mathbb{E}_{(s,a) \sim d_P^\pi(\cdot,\cdot|s_0,a_0)}[Q_{\widehat{P}}^\pi(s, a)]. \tag{58}$$

(57)+(58):

$$Q_P^\pi(s_0, a_0) - Q_{\widehat{P}}^\pi(s_0, a_0) = \frac{\gamma}{1 - \gamma} \mathbb{E}_{s \sim P(\cdot|s_0,a_0), a \sim \pi(\cdot|s)} \left[ \mathbb{E}_{(s',a') \sim d_P^\pi(\cdot,\cdot|s,a)} Q_{\widehat{P}}^\pi(s', a') \right]$$
$$- \frac{\gamma}{1 - \gamma} \mathbb{E}_{(s,a) \sim d_P^\pi(\cdot,\cdot|s_0,a_0)} \left[ \mathbb{E}_{s' \sim \widehat{P}(\cdot|s,a), a' \sim \pi(\cdot|s')} Q_{\widehat{P}}^\pi(s', a') \right]. \tag{59}$$

Consider the first term on right hand side:

$$\mathbb{E}_{s \sim P(\cdot|s_0,a_0), a \sim \pi(\cdot|s)} \mathbb{E}_{(s',a') \sim d_P^\pi(\cdot,\cdot|s,a)}[\cdot] = \mathbb{E}_{(s',a') \sim \widetilde{d}_P^\pi(\cdot,\cdot|s_0,a_0)}[\cdot]$$
$$= \mathbb{E}_{(s,a) \sim d_P^\pi(\cdot,\cdot|s_0,a_0)} \mathbb{E}_{s' \sim P(\cdot|s,a), a' \sim \pi(\cdot|s')}[\cdot]$$

where $\widetilde{d}_P^\pi(s, a|s_0, a_0) := (1 - \gamma) \sum_{t=0}^\infty \gamma^t P^\pi(S_{t+1} = s, A_{t+1} = a|S_0 = s_0, A_0 = a_0)$.

So (59) can be rewritten as

$$Q_P^\pi(s_0, a_0) - Q_{\widehat{P}}^\pi(s_0, a_0)$$
$$= \frac{\gamma}{1-\gamma} \mathbb{E}_{(s,a) \sim d_P^\pi(\cdot,\cdot|s_0,a_0)} \left[ \mathbb{E}_{s' \sim P(\cdot|s,a), a' \sim \pi(\cdot|s')} Q_{\widehat{P}}^\pi(s', a') - \mathbb{E}_{s' \sim \widehat{P}(\cdot|s,a), a' \sim \pi(\cdot|s')} Q_{\widehat{P}}^\pi(s', a') \right]$$
$$= \frac{\gamma}{1-\gamma} \mathbb{E}_{(s,a) \sim d_P^\pi(\cdot,\cdot|s_0,a_0)} \left[ \mathbb{E}_{s' \sim P(\cdot|s,a)} V_{\widehat{P}}^\pi(s') - \mathbb{E}_{s' \sim \widehat{P}(\cdot|s,a)} V_{\widehat{P}}^\pi(s') \right].$$

Finally, consider $V_P^\pi(s_0)$, $V_{\widehat{P}}^\pi(s_0)$ and the initial distribution $\mu$. Recall that $d_P^\pi$ is the visitation measure conditioning on the initial distribution $\mu$. So we have

$$V_P^\pi - V_{\widehat{P}}^\pi = \mathbb{E}_{s_0 \sim \mu} \left[ V_P^\pi(s_0) - V_{\widehat{P}}^\pi(s_0) \right]$$
$$= \mathbb{E}_{s_0 \sim \mu, a_0 \sim \pi(\cdot|s_0)} \left[ Q_P^\pi(s_0, a_0) - Q_{\widehat{P}}^\pi(s_0, a_0) \right]$$
$$= \frac{\gamma}{1-\gamma} \mathbb{E}_{s_0 \sim \mu, a_0 \sim \pi(\cdot|s_0)} \mathbb{E}_{(s,a) \sim d_P^\pi(\cdot,\cdot|s_0,a_0)} \left[ \mathbb{E}_{s' \sim P(\cdot|s,a)} V_{\widehat{P}}^\pi(s') - \mathbb{E}_{s' \sim \widehat{P}(\cdot|s,a)} V_{\widehat{P}}^\pi(s') \right]$$
$$= \frac{\gamma}{1-\gamma} \mathbb{E}_{(s,a) \sim d_P^\pi} \left[ \mathbb{E}_{s' \sim P(\cdot|s,a)} V_{\widehat{P}}^\pi(s') - \mathbb{E}_{s' \sim \widehat{P}(\cdot|s,a)} V_{\widehat{P}}^\pi(s') \right],$$

which finishes the first part of the lemma.

Then we prove the second part: first note that

$$\left| V_P^\pi - V_{\widehat{P}}^\pi \right| = \frac{\gamma}{1-\gamma} \left| \mathbb{E}_{(s,a) \sim d_P^\pi} \left[ \mathbb{E}_{s' \sim P(\cdot|s,a)} V_{\widehat{P}}^\pi(s') - \mathbb{E}_{s' \sim \widehat{P}(\cdot|s,a)} V_{\widehat{P}}^\pi(s') \right] \right|$$
$$\leq \frac{\gamma}{1-\gamma} \mathbb{E}_{(s,a) \sim d_P^\pi} \left| \mathbb{E}_{s' \sim P(\cdot|s,a)} V_{\widehat{P}}^\pi(s') - \mathbb{E}_{s' \sim \widehat{P}(\cdot|s,a)} V_{\widehat{P}}^\pi(s') \right|. \tag{60}$$

Suppose $q_1$, $q_2$ are two arbitrary probability distributions, and $C$ is a constant satisfying $-C \leq f(x) \leq C$. By property of total variation distance, $\mathrm{TV}(q_1, q_2) = \frac{1}{2} \|q_1 - q_2\|_1$.

By Hölder inequality

$$|\mathbb{E}_{x \sim q_1} f(x) - \mathbb{E}_{x \sim q_2} f(x)| = \left| \int f(x)(q_1(x) - q_2(x)) dx \right|$$
$$= \|f(q_1 - q_2)\|_1 \leq \|f\|_\infty \|q_1 - q_2\|_1 \leq 2C\mathrm{TV}(q_1, q_2). \tag{61}$$

Apply (61) to the right hand side of (60):

$$\left| V_P^\pi - V_{\widehat{P}}^\pi \right| \leq 2C \frac{\gamma}{1-\gamma} \mathbb{E}_{(s,a) \sim d_P^\pi} \left[ \mathrm{TV}(P(\cdot|s,a), \widehat{P}(\cdot|s,a)) \right],$$

which concludes the second part.

Third part: Consider the special case that $0 \leq f(x) \leq C$, then we can improve the upper bound in (61)

$$|\mathbb{E}_{x\sim q_1}f(x) - \mathbb{E}_{x\sim q_2}f(x)|$$

$$= \left|\int f(x)(q_1(x) - q_2(x))dx\right|$$

$$= \left|\int f(x)(q_1(x) - q_2(x))\mathbf{1}\{q_1(x) > q_2(x)\}dx - \int f(x)(q_2(x) - q_1(x))\mathbf{1}\{q_1(x) \leq q_2(x)\}dx\right|.$$

Note that on the right hand side, the two terms inside the absolute value sign are both non-negative, so

$$|\mathbb{E}_{x\sim q_1}f(x) - \mathbb{E}_{x\sim q_2}f(x)|$$

$$\leq \max\left\{\int f(x)(q_1(x) - q_2(x))\mathbf{1}\{q_1(x) > q_2(x)\}dx, \int f(x)(q_2(x) - q_1(x))\mathbf{1}\{q_1(x) \leq q_2(x)\}dx\right\}$$

$$\leq C\max\left\{\int (q_1(x) - q_2(x))\mathbf{1}\{q_1(x) > q_2(x)\}dx, \int (q_2(x) - q_1(x))\mathbf{1}\{q_1(x) \leq q_2(x)\}dx\right\}$$

$$= C\mathrm{TV}(q_1, q_2),$$

where the last step is an equivalent definition of total variation distance (for two probability distributions). So the factor 2 on the right hand side in (61) can be improved to 1 in this case. $\square$

**Lemma 2.** *If $f$ is strongly convex with modulus $\mu$ and differentiable, i.e.,*

$$f(y) \geq f(x) + \nabla f(x)^T(y - x) + \frac{\mu}{2}\|y - x\|^2, \tag{62}$$

*suppose $g$ is a convex differentiable function, then $f + g$ is a strongly convex function with modulus $\mu$.*

*Proof.* Since $g$ is a convex function, we have

$$g(y) \geq g(x) + \nabla g(x)^T(y - x). \tag{63}$$

Then

$$f(y) + g(y) \geq f(x) + g(x) + (\nabla f(x)^T + \nabla g(x)^T)(y - x) + \frac{\mu}{2}\|y - x\|^2, \tag{64}$$

$\square$

**Theorem 3.** *Theorem 14.20 of (Wainwright, 2019, chap. 14) (Uniform law for Lipschitz cost functions) Given a uniformly 1bounded function class $\mathcal{F}$ that is star-shaped around the population minimizer $f^*$, let $\delta_n^2 \geq \frac{c}{n}$ be any solution to the inequality*

$$\overline{\mathcal{R}}_n(\delta; \mathcal{F}^*) \leq \delta^2.$$

*(a) Suppose that the cost function is $L$-Lipschitz in its first argument. Then we have*

$$\sup_{f\in\mathcal{F}}\frac{|\mathbb{P}_n(\mathcal{L}_f - \mathcal{L}_f) - \mathbb{P}(\mathcal{L}_f - \mathcal{L}_f)|}{\|f - f^*\|_2 + \delta_n} \leq 10L\delta_n$$

*with probability greater than $1 - c_1 e^{-c_2 n\delta_n^2}$.*
*(b) Suppose that the cost function is $L$-Lipschitz and $\gamma$-strongly convex. Then for any function $\widehat{f} \in \mathcal{F}$ such that $\mathbb{P}_n\left(\mathcal{L}_{\widehat{f}} - \mathcal{L}_f\right) \leq 0$, we have*

$$\left\|\widehat{f} - f^*\right\|_2 \leq \left(\frac{20L}{\gamma} + 1\right)\delta_n$$

*and*

$$\mathbb{P}\left(\mathcal{L}_{\widehat{f}} - \mathcal{L}_f\right) \leq 10L\left(\frac{20L}{\gamma} + 2\right)\delta_n^2,$$

*where both inequalities hold with the same probability as in part (a).*

**Lemma 3.** *Corollary 14.22 of (Wainwright, 2019). Given a class of densities satisfying the previous conditions, let $\delta_n$ be any solution to the critical inequality (14.58) such that $\delta_n^2 \geq \left(1 + \frac{b}{v}\right) \frac{1}{n}$. Then the nonparametric density estimate $\widehat{f}$ satisfies the Hellinger bound*

$$H^2\left(\widehat{f} \| f^*\right) \leq c_0 \delta_n^2$$

*with probability greater than $1 - c_1 e^{-c_2 \frac{v}{b+n} n \delta_n^2}$.*

**Lemma 4.** *Lemma B.2 of Ghosal & Van der Vaart (2017). For every $b > 0$, there exists a constant $\epsilon_b > 0$ such that for all probability densities $p$ and densities $q$ with $0 < d_H^2(p, q) < \epsilon_b P(p/q)^b$,*

$$K(p;q) \lesssim d_H^2(p, q)\left(1 + \frac{1}{b}\log_- d_H(p, q) + \frac{1}{b}\log_+ P\left(\frac{p}{q}\right)^b\right) + 1 - Q(\mathfrak{X}),$$

$$V_2(p;q) \lesssim d_H^2(p, q)\left(1 + \frac{1}{b}\log_- d_H(p, q) + \frac{1}{b}\log_+ P\left(\frac{p}{q}\right)^b\right)^2.$$

*Furthermore, for every pair of probability densities $p$ and $q$ and any $0 < \epsilon < 0.4$,*

$$K(p;q) \leq d_H^2(p, q)\left(1 + 2\log_- \epsilon\right) + 2P\left[\left(\log\frac{p}{q}\right)\mathbb{I}\{q/p \leq \epsilon\}\right],$$

$$V_2(p;q) \leq d_H^2(p, q)\left(12 + 2\log_-^2 \epsilon\right) + 8P\left[\left(\log\frac{p}{q}\right)^2 \mathbb{I}\{q/p \leq \epsilon\}\right].$$

*Consequently, for every pair of probability densities $p$ and $q$,*

$$K(p;q) \lesssim d_H^2(p, q)\left(1 + \log\left\|\frac{p}{q}\right\|_\infty\right) \leq 2d_H^2(p, q)\left\|\frac{p}{q}\right\|_\infty,$$

$$V_2(p;q) \lesssim d_H^2(p, q)\left(1 + \log\left\|\frac{p}{q}\right\|_\infty\right)^2 \leq 2d_H^2(p, q)\left\|\frac{p}{q}\right\|_\infty.$$

## E  Supporting algorithms

Monte Carlo algorithms 4, 5 for evaluating $\tilde{Q}_{\omega,t}(s, A_i)$ at $t$ for each $i = 1, ..., m$ and $\tilde{Q}_{\omega,t}(s, a)$ are provided.

## F  Additional results and details for the numerical studies

### F.1  Synthetic dataset: an illustration

**Environment and behavioral policy details**  For each episode that starts with an initial state $s_0 \sim \mathcal{U}(-2, 2)$, at time $n$ a particle undergoes a random walk and transits according to a mixture of Gaussian dynamics: $s_{n+1} - s_n =: \Delta s \sim \psi_a \mathcal{N}(\mu_{1,a}, 0.1) + (1 - \psi_a)\mathcal{N}(\mu_{2,a}, 0.1)$, where the discrete action $a \in \{-1, 0, 1\}$ corresponds to Left, Stay, and Right, respectively. We choose the random walk steps $\mu_{1,-1} = -2, \mu_{2,-1} = 0$, $\mu_{1,0} = \mu_{1,1} = 0$, $\mu_{2,0} = \mu_{2,1} = 2$ as known parameters, and $\psi_{-1} = \psi_0 = 0.6$, $\psi_1 = 0.4$ as the ground truth unknown model parameters that we estimate with expectation maximization (EM). We generate a partially covered offline dataset collected by a biased (to the left) behavioral policy $\beta$, and define a goal-reaching reward function, respectively given by:

$$\beta(a|s) = \begin{cases} 0.05 & a = 1, \\ 0.05 & a = 0, \\ 0.9 & a = -1, \end{cases} \quad r(s') = \begin{cases} -2 & -3 \leq s' \leq 0, \\ -1.8 & 3 > s' > 0, \\ 0 & s' < -3, \\ 0 & s' \geq 3, \end{cases}$$

---

**Algorithm 4** A Monte Carlo algorithm for evaluating $\tilde{Q}_{\omega,t}(s, A_i)$ at $t$

---

**Input:** The parametric function $f_{t-1,i}(s; \widehat{\beta}_{t-1,i})$.
**Initialization:** Let $s_0 = s$, $a_0 = A_i$, $h = 0$, and $q = r(s_0, A_i)$
**while** TRUE **do**
    Generate $U \sim \text{unif}[0, 1]$.
    **if** $U < 1 - \gamma$ **then**
        Break.
    **else**
        Sample $s_h \sim P_t(\cdot \mid s_{h-1}, a_{h-1})$.
        Solve

$$\pi_t(s_h) = \arg\max_{p' \in \Delta(\mathcal{A})} \left\{ \sum_{i=1}^{m} f_{t-1,i}(s; \widehat{\beta}_{t-1,i}) p'_i - \frac{1}{\eta_t} \omega(p') \right\} \tag{65}$$

        Generate $a_h \sim \pi_t(s_h)$.
        $q = q + r(s_h, a_h)$.
        $h = h + 1$.
    **end if**
**end while**
Let $\widehat{Q}_{P_t}^{\pi_t}(s, A_i) := q$.
Let $\tilde{Q}_{\omega,t}(s, A_i) := \widehat{Q}_{P_t}^{\pi_t}(s, A_i) + \frac{1}{\eta_t} \nabla\omega\left(\pi_t(s)\right)_i$

---

**Algorithm 5** A Monte Carlo algorithm for evaluating $\tilde{Q}_{\omega,t}(s, a)$ for the continuous-action settings

---

**Input:** The parametric function $f_{t-1}(s, a; \widehat{\beta}_{t-1})$.
**Initialization:** Let $s_0 = s$, $a_0 = a$, $h = 0$, and $q = r(s_0, a_0)$
**while** TRUE **do**
    Generate $U \sim \text{unif}[0, 1]$.
    **if** $U < 1 - \gamma$ **then**
        Break.
    **else**
        Sample $s_h \sim P_t(\cdot \mid s_{h-1}, a_{h-1})$.
        Solve

$$\pi_t(s_h) = \arg\max_{a' \in \mathcal{A}} \left\{ f_{t-1}(s, a'; \widehat{\beta}_{t-1}) - \frac{1}{\eta_t} \omega(a') \right\} \tag{66}$$

        Generate $a_h \sim \pi_t(s_h)$.
        $q = q + r(s_h, a_h)$.
        $h = h + 1$.
    **end if**
**end while**
Let $\widehat{Q}_{P_t}^{\pi_t}(s, a) := q$.
Let $\tilde{Q}_{\omega,t}(s, a) := \widehat{Q}_{P_t}^{\pi_t}(s, a) + \frac{1}{\eta_t} \nabla\omega\left(\pi_t(s)\right)_i$

---

for all $s \in \mathbb{R}$. Thus, the particle is encouraged to reach either the positive or negative terminal state with the shortest path possible, with a slight favor towards the positive end if the particle starts off near 0. The offline dataset contains 50 episodes, which are sufficient for an accurate estimation of $\psi_{-1}$ but may lead to misestimation of $\psi_0$ and $\psi_1$. Indeed, for our particular dataset, while the MLE $\hat{\psi}_{-1}$ is accurate, $\hat{\psi}_0$ is underestimated and $\hat{\psi}_1$ is overestimated, which could make over-exploitation of the MLE a problem.

**Implementation details**  We implement MoMA strictly following Algorithm 2. We choose $D(\cdot, \cdot)$ to be KL divergence, reducing the policy improvement steps to natural policy gradient (NPG) as mentioned in

Section 3.2. We parameterize by $f_{t,i}(s, \beta_{t,i}) = \beta_{t,i}^\top e(s, A_i), \forall i = 1, ..., m$ where the features $e(s, A_i)$ are chosen to be exponential functions.

**Contribution from pessimism: accompanying figure**  The accompanying figure referenced in the study of **Contribution from Pessimism** in Section 6.1 is given in Figure 2.

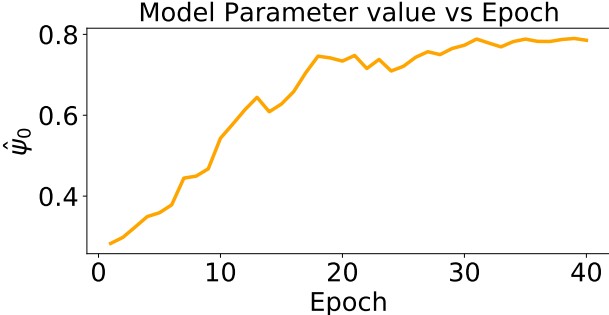

Figure 2: The estimated model parameter $\hat{\psi}_0$ modified by the pessimism updates. As a result, MoMA policy shifts away from the faulty action Stay.

**Hyperparameters and compute information**  The hyperparameters of MoMA used in the random walk experiment are summarized in table 3. The entire run (training and evaluation) of MoMA on a standard CPU takes less than one hour.

Table 3: Hyperparameters for the random walk experiment

| Hyperparameter | Value |
|---|---|
| actor steps | 150 |
| model steps | 150 |
| $\eta$ | 0.1 |
| $\kappa_1$ | 0.1 |
| $\lambda$ | 3.0 |
| MC number | 300 |
| $\gamma$ | 0.4 |
| iterations | 40 |

## F.2   The advantage of nonparametric policy class

We provide an illustrative example to empirically validate the importance of the nonparametric policy class, demonstrating scenarios where the limitations of parametric policies are evident.

Consider an MDP with states $S = \{1, 2, 3, 4, 5\}$ and an action space $A = \{0, 1\}$. The transition dynamics are defined as follows:

$$P(s' = 2 \mid s = 1, a = 0) = 0.8, \quad P(s' = 1 \mid s = 1, a = 0) = 0.2,$$
$$P(s' = 3 \mid s = 1, a = 1) = 0.9, \quad P(s' = 1 \mid s = 1, a = 1) = 0.1,$$
$$P(s' = k \mid s = k, a) = 1 \quad \text{for all } k = 2, 3, 4, 5 \text{ and } a = 0, 1.$$

The reward function is set as follows:

$$
\begin{aligned}
r(s = 1, a = 1) &= 10, \\
r(s = 1, a = 0) &= 5, \\
r(s = 2, a = 1) &= 10, \\
r(s = 2, a = 0) &= 1, \\
r(s, a) &= \sin(s) + \cos(a) \quad \text{for all } s = 3, 4, 5 \text{ and } a = 0, 1.
\end{aligned}
$$

The discount factor $\gamma$ is set to 0.9. Under these settings, the optimal policy is $\pi^*(s = 1) = 1$, $\pi^*(s = 2) = 1$, and $\pi^*(s) = 0$ for all $s = 3, 4, 5$.

We aim to compare the results of the proposed nonparametric policy class and the log-linear policy class, defined as:

$$
\begin{aligned}
\pi(a = 0 \mid s) &= \frac{1}{1 + e^{-\theta_0 + \theta_1 s}}, \\
\pi(a = 1 \mid s) &= 1 - \pi(a = 0 \mid s).
\end{aligned}
$$

Based on these settings, we applied both a parametric policy gradient method and the proposed nonparametric method. The results indicate that the optimal parametric policy yields a sub-optimal policy with a value of 23.50, whereas the proposed nonparametric method successfully identifies the optimal policy with a value of 27.08. This demonstrates the superiority of the nonparametric policy method over a pre-specified policy class.

### F.3 Continuous action D4RL benchmark experiments

**Hyperparameters and compute information** The hyperparameters of MoMA used in the D4RL experiments are summarized in Table 4. We train and evaluate MoMA as well as baseline algorithms on one A100 GPU for all D4RL experiments, and summarize the wall-clock times in Table 5.

Table 4: Hyperparameters for the D4RL experiments

| Hyperparameter | Value |
|---|---|
| $\eta$ | 3E-4 |
| $\kappa_1$ | 3E-4 |
| $\lambda$ | 5E-5 |
| $\gamma$ | 0.99 |

Table 5: D4RL benchmark wall-clock times, rounded to hours.

| | MoMA | MOPO | IQL | CQL |
|---|---|---|---|---|
| Hopper, medium | 6.9 | 2.5 | 4.0 | 6.1 |
| Hopper, medium-replay | 8.1 | 3.2 | 2.4 | 5.5 |
| Hopper, medium-expert | 6.9 | 2.5 | 2.3 | 5.5 |
| HalfCheetah,medium | 8.7 | 4.4 | 5.6 | 8.1 |
| HalfCheetah, medium-replay | 8.7 | 4.4 | 3.2 | 8.1 |
| HalfCheetah, medium-expert | 9.0 | 4.5 | 4.3 | 8.3 |
| Walker2D medium | 10.3 | 2.7 | 2.8 | 6.4 |
| Walker2D, medium-replay | 8.6 | 4.4 | 2.3 | 5.9 |
| Walker2D, medium-expert | 8.9 | 3.2 | 3.4 | 6.0 |

**Alternative evaluation metrics** As a supplement to the standard mean of evaluation scores reported in Table 2, we further consider the evaluation scheme proposed by Agarwal et al. (2021). Their method addresses the need for more reliable performance evaluation in deep reinforcement learning, particularly in

scenarios with limited seeds. It emphasizes the use of distributional metrics, which are less sensitive to outliers, providing a clearer and more robust picture of algorithm performance across different runs.

Specifically, for our MoMA, model-based baseline MOPO, and model-free baseline CQL, we plot the aggregate metrics including interquartile mean (IQM), mean, and optimality gap together with 95% bootstrap confidence intervals (CIs) for each one of the 9 tasks in Figure 3. IQM has better statistical efficiency than median, while optimality gap is a robust alternative to mean. Higher IQM and mean scores are better, and lower optimality gap score is better. We further supplement the aggregate metrics with performance profiles based on score distributions (Agarwal et al., 2021), defined as the fraction of runs above a certain score threshold, and higher curve is better. We plot the performance profiles and bootstrap 95% confidence bands in Figure 4. For the HalfCheetah environment, MoMA consistently outperforms the two baselines across the three data settings measured by all three metrics, and exhibit uniformly higher performance profiles. For the medium-expert data setting, MoMA achieves the best or competitive results across the three environments.

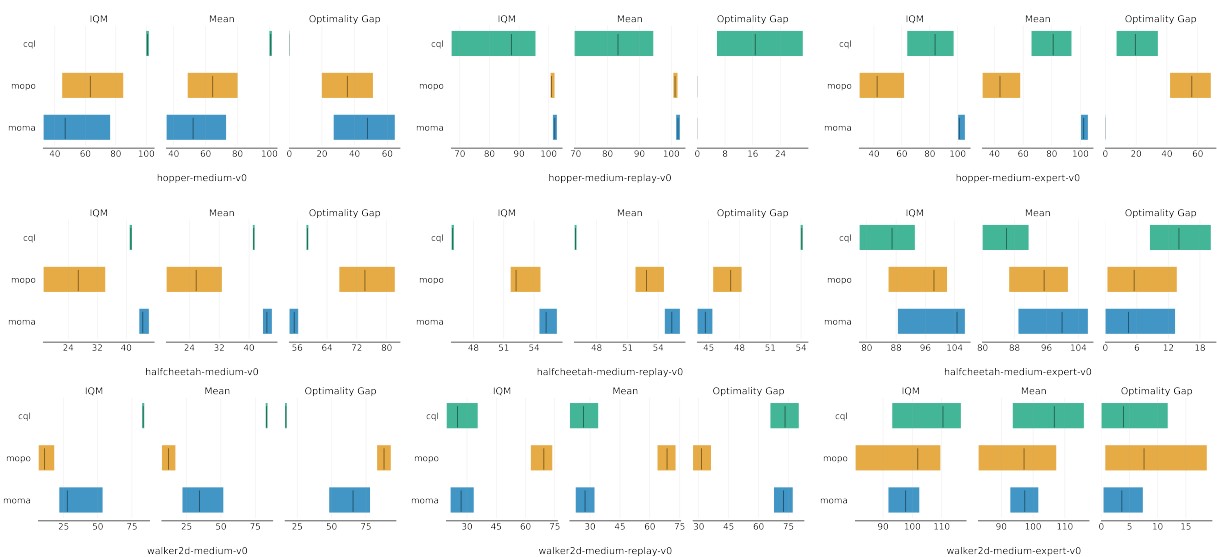

Figure 3: Aggregate metrics on D4RL tasks with 95% bootstrap CIs.

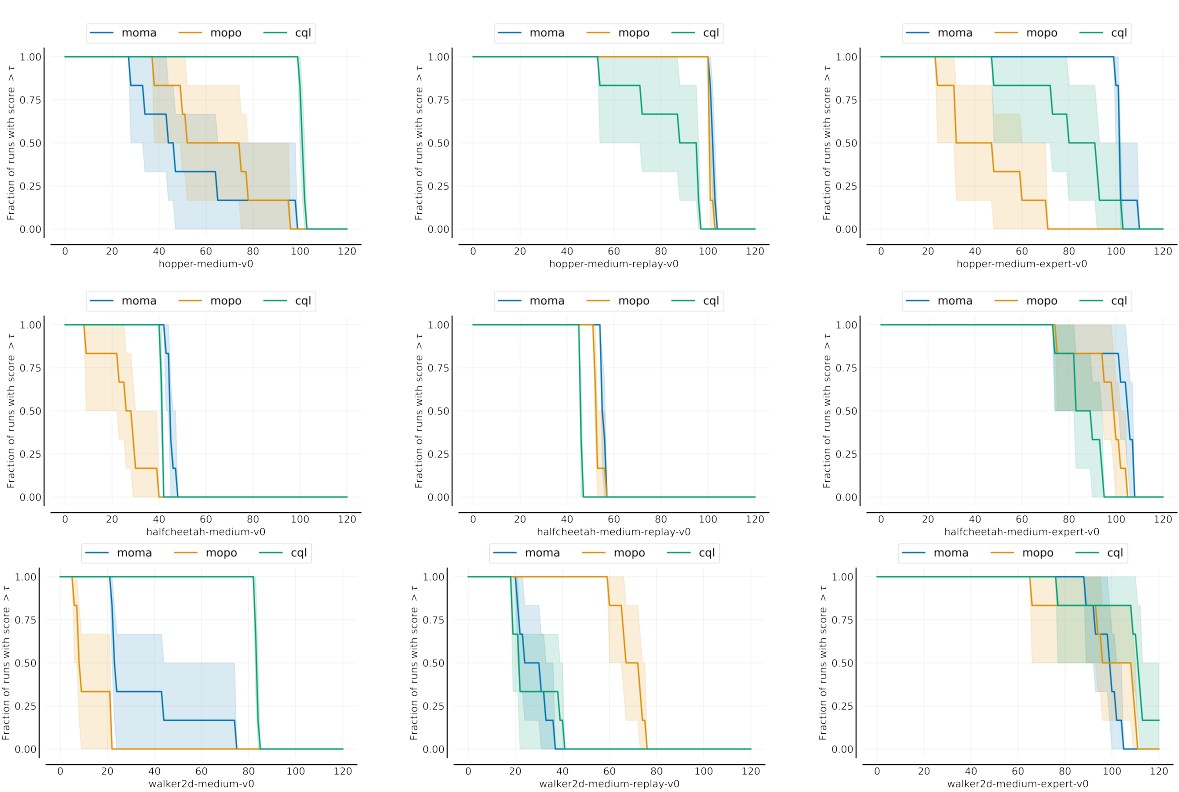

Figure 4: Performance profiles based on score distributions, pointwise 95% confidence bands.

