# OpenReview forum: "MoMA: Model-based Mirror Ascent for Offline Reinforcement Learning"
_TMLR — Accepted by TMLR_

### Review · Reviewer_FipA · 2024-03-11

**Summary Of Contributions:**

The article proposes a new method for off-line Reinforcement Learning based on a model-based conservative estimate of policies (given the offline data available), followed by a policy optimization step based on these conservative estimates. The article starts from a theoretical motivation and introduction of the work, then progresses up to a practical implementation that works with continuous states and actions, evaluated on the D4RL benchmark. These numerical experiments demonstrate that the proposed method almost matches the state of the art.

**Audience:**

Yes

**Broader Impact Concerns:**

The paper does not raise broader impact concerns.

**Claims And Evidence:**

Yes

**Requested Changes:**

1. Critical: improve the clarity of the paper by introducing intuition and writing the paper in a way that the reader does not have to wait for future sections to get an "example" or intuition on how to use a formula
2. Critical: discuss the empirical results to explain why MoMA is still a good algorithm even though it does not match the SOTA in performance. Matching SOTA is not required in a paper (the new RL conference does a great job at explaining why), but here, the paper regularly mentions its superiority to related approaches, and this superiority should either translate to better numerical results, or some other metric being better (compute efficiency, stability, generality).
3. Important but not critical: the paragraph above Equation 5, that discusses the augmented value function, is unreadable. The augmentation should be made much clearer, the notation should be introduced, and ideally examples or intuition should be used.

**Strengths And Weaknesses:**

Strengths:

- The paper has a solid feel and the proposed ideas seem interesting and well-motivated
- The proposed algorithm is very general and seems amenable to many domain-specific variations
- The article goes all the way to a practical implementation and evaluation on the challenging D4RL benchmark.

Weaknesses:

- The clarity of the paper is overall quite low, and the paper seems to have been crammed in the page limit, with, possibly, various authors fighting for space. As a result, some very important parts of the paper are left out (pushed to the Appendix or completely left out), such as the entire page that should be in place of "we adjust the implementation accordingly"
- The order in which things are presented is sub-optimal. The paper is constantly referring to the future, omitting intuition and replacing it with "we will show an example in Section Y".
- In general, there is a lack of intuition or plan behind the formulae. It starts just above Equation 1, with a paragraph that directly starts with notations involving $\mathcal{E}_n(P)$, without giving a definition of these symbols, nor an explanation of where we are going.
- Section 4.3, "An example", should be much larger (and also describe the continous-action setting). It presents several contributions, leverages existing work and is smart about how to compute the gradient of the model, but everything is super-compact and quite unreadable.
- The numerical results are too bad for a discussion of them to be omitted. The proposed algorithm does not match the SOTA, while the paper regularly claims the theoretical superiority of the proposed method. What is the catch? Is the proposed method much more compute-efficient, stable or easy to tune that the related work? Are there easy ways to improve the algorithm to match the SOTA?

---

> ### Author Response · Authors · 2024-06-18
>
> We are grateful for your valuable feedback and appreciate your recognition of the strengths of our paper, including the solid foundation of our proposed ideas and the generality of our algorithm. Below, we would like to address your concerns. All major changes are highlighted in blue in the revised main manuscript and appendix.
>
> > Regarding the comment on clarity listed in weakness 1-3 and requested changes 1
>
> Thank you for your valuable comments. We have revisited the structure of our paper and enhanced its content to ensure that each section adequately conveys its meaning and importance. Specifically, in the revision, we have moved the content from Appendices A and B in the original version into the main text to ensure that critical information is readily accessible. In Section 3, we have expanded on the motivation behind the proposed algorithm's design, providing a clearer rationale for our approach. Additionally, we have provided more intuition and definitions for the symbols and notations used throughout the paper to ensure clarity and better understanding. Furthermore, we have restructured the order of presentation to avoid excessive forward references and to provide examples and intuitive explanations alongside key formulae and concepts. These changes aim to make the paper more cohesive and easier to follow.
>
> > Regarding the comment in weakness 4
>
> In response to your comments on Section 4.3, "An example," we have revised this section to include more detailed explanations. Moreover, we have added the continuous-action setting, detailing the specific computational methods employed in this context. We have also provided a sketch of the proof for the main theorem to further enhance understanding.
>
> > Regarding the comments on numerical experiments listed in the weakness part and critical changes part
>
> Thank you for raising this point. We acknowledge that the numerical results in this work do not always outperform the SOTA. In only two tasks (Hopper-medium-replay and HalfCheetah-medium-expert), our results surpass the SOTA, as shown in Table 2.
>
> The primary contribution of this work lies in the development of a model-based offline RL algorithm without the need for explicit parameterization, along with its theoretical analysis. A notable theoretical advantage of our method is that the size of the policy class does not impact the upper bound of suboptimality, distinguishing our approach from existing offline methods.
>
> While our numerical results do not consistently outperform the SOTA, one plausible reason is that the policy classes used in other offline algorithms might be sufficiently rich for these tasks. Our algorithm would gain more advantage in tasks where the optimal policy is harder to approximate. Furthermore, MoMA is applicable under general function approximation settings, making it a general tool for a wide range of problems.
>
> Improving the algorithm to consistently match the SOTA is a challenging task and an important area for future research. We have added a discussion of these points in the revised manuscript to provide a more comprehensive understanding of MoMA's strengths and potential areas for improvement.
>
> > Regarding the requested change 3.
>
> We have revised the paragraph above Equation 5, which discusses the augmented value function, as per your suggestion.

---

### Review · Reviewer_Pmk3 · 2024-04-30

**Summary Of Contributions:**

The paper presents MoMA (Model-based Mirror Ascent), an offline reinforcement learning algorithm developed to function effectively under partial data coverage without the use of parametric policy spaces. The algorithm operates through two primary steps: a conservative policy evaluation, where the Q function is minimized over a confidence set of transition models to accommodate partial coverage scenarios; and a policy improvement step, which employs mirror ascent with general function approximations, facilitating the use of an unrestricted policy class. This approach is designed to enhance the algorithm's applicability in practical settings.

From a theoretical perspective, the authors provide an upper bound on the suboptimality of MoMA, noting that it does not incorporate the size of the policy class, which may allow for broader application compared to model-free methods. The paper also details a practically implementable version of MoMA, including a primal-dual step designed to approximately solve the constrained minimization problem. The efficacy of MoMA is evaluated through numerical studies using both synthetic data and the D4RL benchmark.

**Audience:**

Yes

**Claims And Evidence:**

Yes

**Requested Changes:**

The paper already represents a valuable addition to the literature on offline reinforcement learning. To strengthen the submission further, the following revisions are suggested:

1. Demonstration of non-parametric policy benefits: The theoretical benefit of MoMA operating independently of the policy class parameterization is highlighted but not empirically validated. It would be beneficial if the authors could include experiments in settings where the limitations of parametric policies are evident, or demonstrate situations where the optimal policy would likely fall outside a typical parametric class. This would substantiate the theoretical advantages discussed.

2. Reevaluation of synthetic data results: The overlapping confidence intervals reported in Table 1 suggest that the differences in performance between MoMA and other methods may not be statistically significant. A revision of the experimental design, perhaps by introducing a more challenging synthetic dataset, could help in demonstrating MoMA's effectiveness more convincingly. This might involve designing experiments that more clearly showcase the algorithm's strengths in controlled settings.

**Strengths And Weaknesses:**

__Strengths__

* Novel algorithm design that combines conservative policy evaluation --- involving solving for a constrained model optimization, followed by a policy improvement step using mirror ascent.
* Detailed theoretical exposition
* Practical implementation and ablation insights: the implementation of MoMA is complemented by numerical experiments on D4RL continuous control tasks, showcasing its effectiveness. Additionally, Figure 1 provides interesting ablative insights into MoMA's approach to value conservatism, compared to NPG.

__Weaknesses__

* Lack of empirical validation for theoretical claims: The paper does not empirically demonstrate the claimed benefits of MoMA’s independence from the policy class parameterization. The environments used, including those from D4RL, do not necessitate non-parametric policies, thus not substantiating the theoretical advantage.
* Limited Value of Synthetic Data Experiment: The synthetic data experiments show overlapping confidence intervals among competing methods in Table 1, suggesting that MoMA’s advantages may not be statistically significant in controlled environments.

---

> ### Author Response · Authors · 2024-06-18
>
> We sincerely appreciate your constructive feedback. We are willing to address each of your queries in this response.
>
> >Regarding the comment on demonstration of non-parametric policy benefits listed in Weakness 1 and requested changes 1
>
> We appreciate your feedback on empirically validating the theoretical benefit of MoMA's non-parametric approach. We are currently designing experiments to compare algorithms that use explicit parameterized policy classes (such as log-linear policy classes) with those that do not require explicit policy parameterization. Specifically, we employ reproducing kernel Hilbert space (RKHS) to model the augmented value function, a well-known non-parametric method. Due to time constraints, we have not yet obtained the numerical results. However, we are making efforts to update the numerical results before the deadline of this rebuttal period. Thank you for your understanding and consideration.
>
>
> >Regarding the comment on limited value of synthetic data experiment listed in Weakness 2 and requested changes 2
>
> Thank you. Following your suggestion, we have designed a more complex synthetic dataset and are currently conducting numerical experiments on this dataset. Specifically, instead of the original one-dimensional state space with a simple transition kernel, we are now considering a multi-dimensional state space with a more complex transition kernel. Additionally, we will replicate the experiment using more random seeds to improve the accuracy of the confidence intervals for the estimated policy values. Again, we are making efforts to update the numerical results before the deadline of this rebuttal period.

---

> ### Author Response · Authors · 2024-06-24
>
> We would like to provide an update regarding the numerical experiments to address your concerns about the benefits of the nonparametric policy class. We have used an illustrative example to demonstrate the importance of the nonparametric policy class.
>
> We consider an MDP with states \( S = \{1,2,3,4,5\} \) and action space \( A = \{0,1\} \). The transition dynamics are defined as follows:
> \begin{align*}
>     &P(s' = 2 \mid s = 1, a = 0) = 0.8, \quad P(s' = 1 \mid s = 1, a = 0) = 0.2, \\
>     &P(s' = 3 \mid s = 1, a = 1) = 0.9, \quad P(s' = 1 \mid s = 1, a = 1) = 0.1, \\
>     &P(s' = k \mid s = k, a) = 1 \quad \text{for all } k = 2, 3, 4, 5 \text{ and } a = 0, 1.
> \end{align*}
>
> The reward function is set as follows:
> \begin{align*}
>     &r(s = 1, a = 1) = 10, \\
>     &r(s = 1, a = 0) = 5, \\
>     &r(s = 2, a = 1) = 10, \\
>     &r(s = 2, a = 0) = 1, \\
>     &r(s, a) = \sin(s) + \cos(a) \quad \text{for all } s = 3, 4, 5 \text{ and } a = 0, 1.
> \end{align*}
>
> The discount factor \( \gamma \) is set to 0.9. Under these settings, the optimal policy is \\( \pi^*(s = 1) = 1 \\), \\( \pi^*(s = 2) = 1 \\), and \\( \pi^*(s) = 0 \\) for all \( s = 3, 4, 5 \).
>
> We aim to compare the results of the proposed nonparametric policy class and the log-linear policy class, defined as:
> \begin{align*}
>     &\pi(a = 0 \mid s) = \frac{1}{1 + e^{-\theta_0 + \theta_1 s}}, \\
>     &\pi(a = 1 \mid s) = 1 - \pi(a = 0 \mid s).
> \end{align*}
>
> Based on these settings, we performed a parametric policy gradient method and the proposed nonparametric method. The results show that the optimal parametric policy returns a sub-optimal policy with a policy value of 23.50, while the proposed nonparametric method is able to find the optimal policy with a policy value of 27.08. This demonstrates the advantage of the nonparametric policy method compared to a pre-specified policy class.
>
> We have added a discussion in Appendix H.3.

---

### Review · Reviewer_VumW · 2024-06-09

**Summary Of Contributions:**

The paper proposes a MoMA algorithm for offline policy training. MoMA works iteratively, with each iteration comprised of two stages. In the first stage (conservative policy evaluation), an MDP that minimizes the value function of a current policy is chosen among all plausible MDPs. In the second stage (policy improvement), the next step policy is derived as a maximizer of an expected current $Q$-value under a (soft) constraint penalizing the distance from the current policy. The paper includes theoretical and experimental results.

**Audience:**

Yes

**Claims And Evidence:**

No

**Requested Changes:**

See above.

**Strengths And Weaknesses:**

Strengths:
* MoMA is a blend of two historically justified RL ideas: (a) picking a conservative estimate of V using a pessimistic MDP across plausible MDPs and (b) finding a policy that maximizes a regularized expected sum of rewards.
* The paper claims that it has a better expressiveness of the policy class and more flexibility of function approximators (though see discussion below).
* Theoretical analysis is provided.

Weaknesses:
* General
	* The paper includes long proofs (in Appendix). It would be useful to:
		* Provide a sketch of the main argument in the paper's body.
		* Compare the complexity results obtained in this paper with the known facts from the literature (e.g., in the form of a table).
	* Slightly related to the previous item, the content of Appendix A is interesting and could be included in the main body.
	* The paper claims two strengths of MoMA in the form of (a) unrestricted policy class (quote from Section 3.1: "has no restriction on the policy class, which is crucial when the optimal policy is not contained in a restricted parametric policy class"), and (b) policy updates with general function approximations (quote from Section 3.2 "the update rule [...] does not require any explicit policy parameterization"). However, an approximate version of the algorithm is used for the experiments with neural networks and SGD optimization. Additionally, the results in the experiments section are not as convincing as they are claimed to be in the paper (see below).
* Experiments:
	* The experiments have a small number of seeds (5).
	* The confidence intervals are misleading. First, only five seeds are used for the computations. Second, the CIs are constructed as the mean +/- std of the result. Such an approach is usually done with the asymptotic assumption (which is controversial given the number of seeds), in which case one assumes the CI form for the normal distribution. In this case, however, +/- std corresponds to a confidence level $\approx 68\%$, an uncommonly low figure. As a side-note, for the general case with a finite std, one could hope to use the Chebyshev's inequality, but it is uninformative in the case at hand, giving a trivial upper-bound of $100\%$.
	* The above leads to artificially narrowed down CIs and, consequently, to overoptimistic interpretation of the results.
	* On page 9, the paper claims "notably, in 4 out of 9 cases, our algorithm outperforms the other three model-based RL algorithms" (referring to Table 2).
		* There are two model-based baselines.
		* Taking into account the overlap between confidence intervals (which are misleadingly narrow) MoMA significantly beats **model-based** baselines only in one case (Hopper, medium-expert). However, it does not significantly beat all baselines (model-based and model-free combined).
	* There is no comparison of wall time or compute between the methods (only how long did the MoMA training and eval take in Appendix H2)
* Other remarks:
	* Undefined interval $\Lambda$ (below equation (4)).
	* The form of $\omega$ function assumed in the experiments section should be made clear.
	* Section 4.2 is dense with multiple inline formulas, which is hard to read and not immediately easy to follow.
	* For random walk, $\gamma=0.4$ (Appendix H.1.), which seems to be unusually low.

---

> ### Author Response · Authors · 2024-06-18
>
> Thank you for your constructive feedback. We appreciate the opportunity to address each of your concerns. All major changes are highlighted in blue in the revised main manuscript and appendix.
>
> > Regarding Weakness 1, 2
>
> Thank you for your suggestion. We have provided a sketch of the proof for Theorem 1 and included the content of Appendix A in the main body to ensure that critical information is readily accessible. Additionally, we have included examples and intuitive explanations alongside key formulae and concepts to make the paper more cohesive and easier to follow.
>
> While the primary contributions of our work are not focused on deriving optimal sample complexity, we appreciate the importance of contextualizing our findings within the existing literature. In response to your feedback, we have included a direct comparison with some existing works in the newly-added Section 7 of the revised manuscript, which was originally presented in the Appendix. Specifically, we demonstrate that the suboptimality scales with $\frac{1}{\sqrt{n}}$, outperforming existing model-free works such as [Xie et al.]. Additionally, we highlight that our policy improvement step is computationally efficient under general function approximation settings, unlike the approaches in [Zanette et al.], [Xie et al.], and [Cheng et al.]. We believe these comparisons effectively illustrate the strengths of our method.
>
> > Regarding numerical results
>
> Thank you for your valuable feedback.
>
> We agree with your assessment regarding the limited number of seeds used in our experiments. Due to the computationally intensive nature of the deep learning algorithms, we only ran a small number of seeds. We also acknowledge the resulting limitations reporting confidence intervals (CIs) based on a small number of seeds.
>
> To address your concern, we have revised our evaluation approach in the updated manuscript. We have incorporated the evaluation scheme proposed by Agarwal et al. (2021) to provide more robust and informative metrics. Agarwal et al.'s method is motivated by the need for more reliable performance evaluation in deep reinforcement learning, especially in scenarios with limited seeds. Their approach emphasizes the use of distributional metrics that are less sensitive to outliers and provide a clearer picture of algorithm performance across different runs.
>
> Specifically, we now include the interquartile mean (IQM) and the optimality gap as aggregate metrics, along with 95\% bootstrap CIs. The IQM provides a robust measure of central tendency by focusing on the middle 50\% of the data, reducing the influence of extreme values. The optimality gap measures how close the performance is to an optimal policy, offering a meaningful interpretation of results. Additionally, we present performance profiles based on score distributions, which offer a comprehensive view of the algorithm's performance across different scenarios. These changes have been added to Section 6 of the revised main manuscript and detailed in Appendix F.2.
>
> In Appendix F.2, we have also provided a full table of wall-clock runtime for each algorithm under each task.
>
> > "Undefined interval $\Lambda$ (below equation (4))."
>
> We have added a description of the interval $\Lambda$ in the revision.
>
> > "The form of $\omega$ function assumed in the experiments section should be made clear."
>
> Thank you. We have revised it.
>
> > "Section 4.2 is dense with multiple inline formulas, which is hard to read and not immediately easy to follow."
>
> Thank you for your comment. We have revised section 4.2 to make it clearer in the updated version.
>
>
> > "For random walk, $\gamma=0.4$ (Appendix F.1.), which seems to be unusually low."
>
> Thank you for your observation. We have experimented with different values of $\gamma$, and the numerical results indicate that $\gamma=0.4$ is a sufficient choice to demonstrate the efficacy of the proposed method in our simulation study on the synthetic dataset.
>
> [1] Tengyang Xie, Ching-An Cheng, Nan Jiang, Paul Mineiro, and Alekh Agarwal. Bellman-consistent pessimism
> for offline reinforcement learning. Advances in neural information processing systems, 34:6683–6694, 2021.
>
> [2] Andrea Zanette, Martin J Wainwright, and Emma Brunskill. Provable benefits of actor-critic methods for
> offline reinforcement learning. Advances in neural information processing systems, 34, 2021.
>
> [3] Ching-An Cheng, Tengyang Xie, Nan Jiang, and Alekh Agarwal. Adversarially trained actor critic for offline
> reinforcement learning. arXiv preprint arXiv:2202.02446, 2022.
>
> [4] Rishabh Agarwal, Max Schwarzer, Pablo Samuel Castro, Aaron C Courville, and Marc Bellemare. Deep
> reinforcement learning at the edge of the statistical precipice. Advances in neural information processing
> systems, 34:29304–29320, 2021.

---

### Decision · Action_Editor_ynpA · 2024-09-06

**Recommendation:** Accept with minor revision

**Comment:**

The paper presents a novel approach to offline reinforcement learning with theoretical foundations and practical implementations. While there are some concerns about the clarity of presentation and the strength of empirical results, the innovative aspects of the work - particularly the use of an unrestricted policy class and the combination of conservative policy evaluation with mirror ascent - make it a contribution to the field.

To address the reviewers' concerns, the authors should be encouraged to:

- Improve the clarity of the paper, especially in explaining theoretical concepts and providing intuition for the algorithms.
- Strengthen the empirical validation, particularly in demonstrating the benefits of the unrestricted policy class.
- Provide a more balanced discussion of the results, acknowledging where MoMA does not outperform existing methods and explaining potential reasons.

These revisions would enhance the paper's impact and usefulness to the TMLR audience without fundamentally altering its core contributions.

**Audience:**

Researchers in reinforcement learning, particularly those focused on offline RL and model-based methods, would likely find this paper of interest. The combination of conservative policy evaluation with mirror ascent policy improvement, along with the theoretical analysis provided, could appeal to those studying RL theory. Additionally, practitioners working on applying RL to continuous control tasks might be intrigued by the practical implementation and D4RL benchmark results, although the mixed performance compared to state-of-the-art methods may temper enthusiasm. While the paper's contributions are noteworthy, the audience's interest might be moderated by the need for clearer exposition and more robust empirical validation of the theoretical claims.

**Claims And Evidence:**

- The paper provides theoretical analysis and proofs to support its claims about MoMA's properties and performance bounds. However, some reviewers felt the theoretical arguments could be better explained or contextualized in the main text.

- The experimental results do not fully support some of the paper's claims about MoMA's superiority:
  - MoMA does not consistently outperform baselines across all environments tested
  - The confidence intervals are narrow and potentially misleading due to the small number of seeds used
  - There is limited empirical validation of the claimed benefits of MoMA's unrestricted policy class

- The synthetic data experiments show overlapping confidence intervals between MoMA and competing methods, suggesting the advantages may not be statistically significant in controlled environments.

- The paper claims theoretical advantages of MoMA's independence from policy class parameterization, but does not empirically demonstrate these benefits in environments that would necessitate non-parametric policies.